# ATTENTION WITH MARKOV: A CURIOUS CASE OF SINGLE-LAYER TRANSFORMERS

**Ashok Vardhan Makkuva**[*]
EPFL

**Marco Bondaschi**[*]
EPFL

**Adway Girish**
EPFL

**Alliot Nagle**
UT Austin

**Martin Jaggi**
EPFL

**Hyeji Kim**[†]
UT Austin

**Michael Gastpar**[†]
EPFL

## ABSTRACT

Attention-based transformers have achieved tremendous success across a variety of disciplines including natural languages. To deepen our understanding of their sequential modeling capabilities, there is a growing interest in using Markov input processes to study them. A key finding is that when trained on first-order Markov chains, transformers with two or more layers consistently develop an induction head mechanism to estimate the in-context bigram conditional distribution. In contrast, single-layer transformers, unable to form an induction head, directly learn the Markov kernel but often face a surprising challenge: they become trapped in local minima representing the unigram distribution, whereas deeper models reliably converge to the ground-truth bigram. While single-layer transformers can theoretically model first-order Markov chains, their empirical failure to learn this simple kernel in practice remains a curious phenomenon. To explain this contrasting behavior of single-layer models, in this paper we introduce a new framework for a principled analysis of transformers via Markov chains. Leveraging our framework, we theoretically characterize the loss landscape of single-layer transformers and show the existence of global minima (bigram) and bad local minima (unigram) contingent on data properties and model architecture. We precisely delineate the regimes under which these local optima occur. Backed by experiments, we demonstrate that our theoretical findings are in congruence with the empirical results. Finally, we outline several open problems in this arena. Code is available at https://github.com/Bond1995/Markov.

## 1 INTRODUCTION

Attention-based transformers have been at the forefront of recent breakthroughs in a variety of disciplines, including natural language processing (Vaswani et al., 2017; Radford and Narasimhan, 2018; Devlin et al., 2018). One of the key workhorses behind this success is the attention mechanism, which allows transformers to capture complex causal structures in the data, thus rendering them with impressive sequential modeling capabilities.

Given their success, there is tremendous interest in understanding the sequential modeling abilities of transformers. Notably, a growing body of research explores transformers through Markov input processes to investigate their in-context learning capabilities (Rajaraman et al., 2024a; Nichani et al., 2024; Edelman et al., 2024; Bietti et al., 2023). These studies reveal an interesting insight that transformers with two or more layers develop an induction head to estimate the in-context bigram conditional distribution when trained on first-order Markov chains. In contrast, single-layer transformers, unable to form an induction head (Olsson et al., 2022), directly learn the Markov kernel. Surprisingly, we empirically find that while deeper models reliably converge to the ground-truth bigram, regardless of initialization, single-layer transformers often get stuck in local minima corresponding to the unigram distribution (Fig. 1). Despite their theoretical ability to model first-order

---

[*]Equal contribution. [†]Equal contribution. Correspondence to: Ashok Vardhan Makkuva, ashok.makkuva@epfl.ch

Markov chains, they sometimes fail to learn this simple kernel in practice. Motivated by this stark contrast in behavior based on depth, and our limited understanding of it, we ask: *Can we systematically characterize the learning capabilities of single-layer transformers with Markovian inputs?*

To address this, in this paper we introduce a new framework for a principled theoretical and empirical analysis of transformers via Markov chains. Leveraging our framework, we characterize the loss landscape of single-layer transformers and prove the existence of bad local minima and global minima corresponding to the unigram and bigram, respectively. We further demonstrate that the presence of these local optima depends on the Markov state switching probabilities and the transformer's weight-tying, and we precisely delineate the regimes under which this happens. Together, our analysis reveals a complex interplay between the data-distributional properties, the transformer architecture, and the final model performance for single-layer transformers with Markov chains, explaining the aforementioned empirical phenomena.

In summary, we make the following contributions:

- We provide a novel framework for a precise theoretical and empirical study of transformers via Markov chains (Sec. 3).

- We characterize the loss landscape of single-layer transformers with first-order Markov chains, highlighting the effect of the data distribution and the model architecture (Sec. 4).

- We show that the Markov switching probabilities and weight-tying play a crucial role in the presence of local optima on loss surface and precisely characterize the said conditions (Thms. 2 and 3).

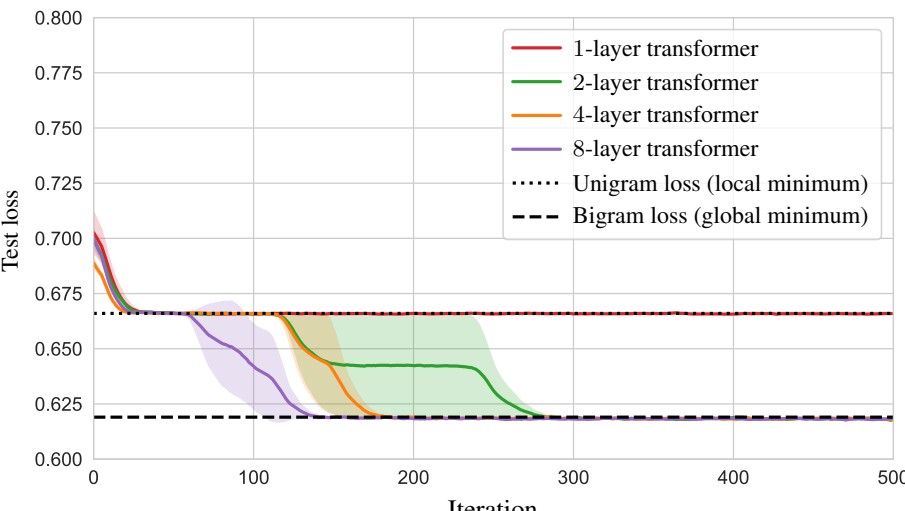

Figure 1: Single-layer transformers get stuck at local minima, corresponding to the unigram model, when the input is a first-order Markov chain with switching probabilities $p = 0.5$ and $q = 0.8$ (Fig. 2b). However, deeper models escape to global minima corresponding to the bigram model.

Our main findings and observations are:

- We prove that weight tying can introduce bad local minima for single-layer transformers when Markovian switching is greater than one (Thm. 2). Removing the tying, however, solves the issue (Thm. 3).

- When the Markovian switching is less than one, we empirically observe the model always converges to the global minima irrespective of the weight tying (Fig. 3).

- Interestingly, transformers with depth two and beyond always converge to the global minima irrespective of the weight tying and switching (Fig. 1).

**Notation.** Scalars are denoted by italic lower case letters like $x, y$ and Euclidean vectors and matrices are denoted by bold ones $\boldsymbol{x}, \boldsymbol{y}, \boldsymbol{M}$, etc. We use $\|\cdot\|$ to denote the $\ell_2$-norm for Euclidean

vectors and Frobenius norm for matrices. $[k] \triangleq \{1, \ldots, k\}$, and for a sequence $(x_n)_{n \geq 1}$, define $x_k^m \triangleq (x_k, \ldots, x_m)$ if $k \geq 1$ and $(x_1, \ldots, x_m)$ otherwise. For $z \in \mathbb{R}$, the sigmoid $\sigma(z) \triangleq 1/(1 + e^{-z})$ and $\mathrm{ReLU}(z) \triangleq \max(0, z)$. For events $A$ and $B$, $\mathbb{P}(A)$ denotes the probability of $A$ whereas $\mathbb{P}(A \mid B)$ the conditional probability. Let $(x, y)$ be a pair of discrete random variables on $[k] \times [k]$ with the probability mass function (pmf) of $x$ being $\boldsymbol{p}_x = (p_1, \ldots, p_k) \in [0, 1]^k$. Then its Shannon entropy is defined as $H(x) = H(\boldsymbol{p}_x) \triangleq -\sum_{i \in [k]} p_i \log p_i$, and the conditional entropy $H(y|x) \triangleq H(x, y) - H(x)$. The entropy rate of a stochastic process $(x_n)_{n \geq 1}$ is defined as $\lim_{n \to \infty} H(x_1^n)/n$. Finally, for $p \in (0, 1)$, the binary entropy function $h(\cdot)$ is defined as $h(p) \triangleq H(p, 1 - p) = -p \log p - (1 - p) \log(1 - p)$.

## 2 BACKGROUND

We describe the transformer architecture and the Markovian input process.

### 2.1 TRANSFORMERS

We study a single-layer transformer with a single-head softmax attention and $\mathrm{ReLU}$ non-linearity. We omit the layer norm since its influence is marginal in the settings we consider (Sec. 4). We consider an input vocabulary $\mathcal{X}$ of finite size $|\mathcal{X}|$. For the ease of exposition, in this paper we mainly focus on $|\mathcal{X}| = 2$, i.e. $\mathcal{X} = \{0, 1\}$, and outline our results for multi-state setting in Sec. 4.3. Let $\{x_n\}_{n=1}^N \in \{0, 1\}^N$ be an input sequence of length $N$. Then for each $n \in [N]$, the transformer operations are mathematically given by (Fig. 2a):

$$\boldsymbol{x}_n = x_n \boldsymbol{e}_1 + (1 - x_n) \boldsymbol{e}_0 + \widetilde{\boldsymbol{p}}_n \in \mathbb{R}^d, \qquad \text{(Embedding)}$$

$$\boldsymbol{y}_n = \boldsymbol{x}_n + \boldsymbol{W}_O \sum_{i \in [n]} \mathrm{att}_{n,i} \cdot \boldsymbol{W}_V \boldsymbol{x}_i \in \mathbb{R}^d, \qquad \text{(Attention)}$$

$$\boldsymbol{z}_n = \boldsymbol{y}_n + \boldsymbol{W}_2 \mathrm{ReLU}(\boldsymbol{W}_1 \boldsymbol{y}_n) \in \mathbb{R}^d, \qquad \text{(FF)}$$

$$\mathrm{logit}_n = \langle \boldsymbol{a}, \boldsymbol{z}_n \rangle + b \qquad \in \mathbb{R}, \qquad \text{(Linear)}$$

$$f_{\bar{\boldsymbol{\theta}}}(x_1^n) \triangleq \mathbb{P}_{\bar{\boldsymbol{\theta}}}(x_{n+1} = 1 \mid x_1^n) = \sigma(\mathrm{logit}_n) \in [0, 1]. \qquad \text{(Prediction)}$$

Here $\bar{\boldsymbol{\theta}} \triangleq (\boldsymbol{e}_1, \boldsymbol{e}_0, \{\widetilde{\boldsymbol{p}}_n\} \ldots, b, \boldsymbol{a})$ denotes the full list of the transformer parameters. $d$ is the embedding dimension, $\boldsymbol{e}_1$ and $\boldsymbol{e}_0$ in $\mathbb{R}^d$ are the token-embeddings corresponding to $x_n = 1$ and $x_n = 0$ respectively, and $\widetilde{\boldsymbol{p}}_n$ is the (trainable) positional encoding. We have matrices $\boldsymbol{W}_O \in \mathbb{R}^{d \times m}$ and $\boldsymbol{W}_V \in \mathbb{R}^{m \times d}$, and the attention weights $\mathrm{att}_{n,i} \in (0, 1)$ are computed using the query and key matrices (§ A). $\boldsymbol{W}_2 \in \mathbb{R}^{d \times r}$ and $\boldsymbol{W}_1 \in \mathbb{R}^{r \times d}$ are the weight matrices in the FF layer, whereas $\boldsymbol{a} \in \mathbb{R}^d$ and $b \in \mathbb{R}$ are the weight and bias parameters for the linear layers. For a multi-layer transformer, we apply the successive attention and feed-forward layers multiple times before the final linear layer. Finally, we compute the probability for the symbol 1 using the logits: $f_{\bar{\boldsymbol{\theta}}}(x_1^n) \triangleq \mathbb{P}_{\bar{\boldsymbol{\theta}}}(x_{n+1} = 1 \mid x_1^n) = \sigma(\mathrm{logit}_n) \in [0, 1]$. Note that a single symbol probability suffices as the vocabulary is binary.

**Loss.** The parameters $\bar{\boldsymbol{\theta}}$ are trained using the next-token prediction loss between the predicted probability $f_{\bar{\boldsymbol{\theta}}}(x_1^n)$ and the corresponding ground truth symbol $x_{n+1}$ across all the positions $n \in [N]$:

$$L(\bar{\boldsymbol{\theta}}) \triangleq -\frac{1}{N} \sum_{n \in [N]} \mathbb{E}_{x_1^{n+1}}[x_{n+1} \cdot \log f_{\bar{\boldsymbol{\theta}}}(x_1^n) + (1 - x_{n+1}) \cdot \log(1 - f_{\bar{\boldsymbol{\theta}}}(x_1^n))], \qquad (1)$$

where the expectation is over the data distribution of the sequence $\{x_n\}_{n=1}^N$. In practice, it is replaced by the empirical averages across the sequences $\{x_n\}_{n=1}^N$ sampled from the corpus, with stochastic optimizers like SGD or Adam (Kingma and Ba, 2015) used to update the model parameters.

### 2.2 MARKOV CHAINS

We model the input as a *first-order Markov chain*, i.e. a Markov chain with (order) memory $m = 1$. For these processes, the next state is influenced only by the current state and none of the past:

$$\boldsymbol{P}_{ij} \triangleq \mathbb{P}(x_{n+1} = j \mid x_n = i) = \mathbb{P}\left(x_{n+1} = j \mid x_n = i, x_1^{n-1} = i_1^{n-1}\right),$$

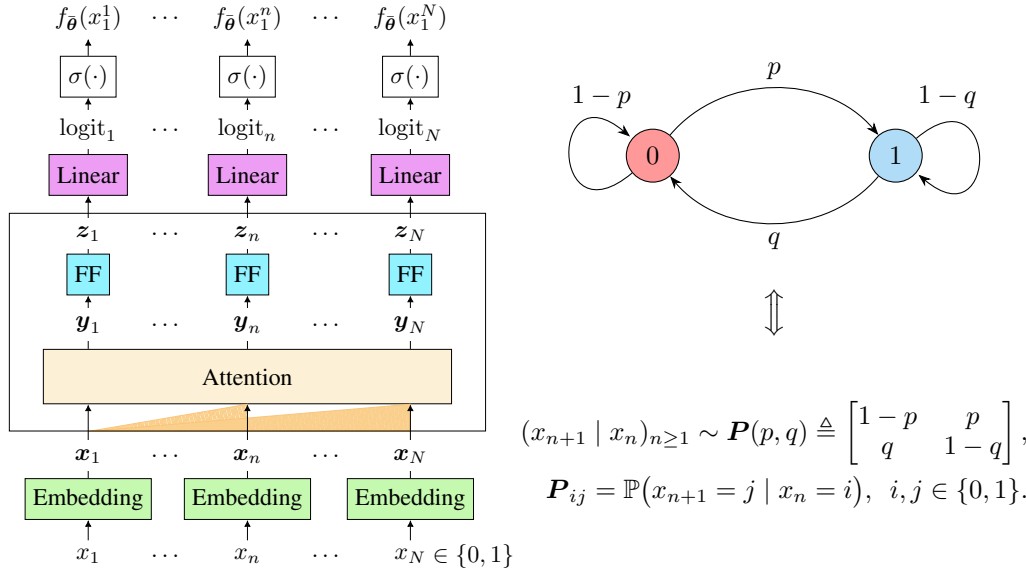

(a) The transformer model with binary input data: for each $x_1^n$, the next-bit prediction probability is $f_{\bar{\boldsymbol{\theta}}}(x_1^n) = \mathbb{P}_{\bar{\boldsymbol{\theta}}}(x_{n+1} = 1 | x_1^n)$.

(b) State transition diagram and Markov kernel for a first-order Markov chain $\boldsymbol{P}(p, q)$ with flipping probabilities $\boldsymbol{P}_{01} = p$ and $\boldsymbol{P}_{10} = q$.

Figure 2: Analysis of transformers via Markov chains.

for any $i_1, \ldots, i_{n-1}, i, j \in \mathcal{X}, n \geq 1$. Here the Markov kernel $\boldsymbol{P} = (\boldsymbol{P}_{ij})$ governs the transition dynamics of the process: if $\boldsymbol{\pi}^{(n)} \in [0, 1]^{|\mathcal{X}|}$ denotes the probability law of $x_n$ at time $n$, then $\boldsymbol{\pi}^{(n+1)} = \boldsymbol{\pi}^{(n)} \cdot \boldsymbol{P}$. Of particular interest to us in this paper is the kernel $\boldsymbol{P}(p, q) \triangleq [1-p, p\,;\, q, 1-q]$ on the binary state space with the switching probabilities $\boldsymbol{P}_{01} = p$ and $\boldsymbol{P}_{10} = q$, for $p, q \in (0, 1)$. Fig. 2b illustrates the state transition diagram for this kernel. Here we refer to the sum $p + q$ as the *switching factor*. We denote a first-order binary Markov chain $(x_n)_{n \geq 1}$ with the transition kernel $\boldsymbol{P}(p, q)$ and starting with an initial law $\boldsymbol{\pi}^{(1)}$ as $(x_n)_{n \geq 1} \sim (\boldsymbol{\pi}^{(1)}, \boldsymbol{P}(p, q))$. When the initial distribution is understood from context, we simply write $(x_{n+1} \mid x_n)_{n \geq 1} \sim \boldsymbol{P}(p, q)$. For this process, the entropy rate equals $H(x_{n+1} | x_n) = \frac{1}{p+q}(q\,h(p) + p\,h(q))$, which is independent of $n$.

**Stationary distribution.** A *stationary distribution* of a Markov chain is a distribution $\boldsymbol{\pi}$ on $\mathcal{X}$ that is invariant to the transition dynamics, i.e. if $\boldsymbol{\pi}^{(n)} = \boldsymbol{\pi}$, then we have $\boldsymbol{\pi}^{(n+1)} = \boldsymbol{\pi}\boldsymbol{P} = \boldsymbol{\pi}$ and consequently, $\boldsymbol{\pi}^{(m)} = \boldsymbol{\pi}$ for all $m \geq n$. Also referred to as the steady-state distribution, its existence and uniqueness can be guaranteed under fairly general conditions (Norris, 1997), and in particular for $\boldsymbol{P}(p, q)$ when $p, q \neq 0, 1$. For $\boldsymbol{P}(p, q)$, the stationary distribution is given by $\boldsymbol{\pi}(p, q) \triangleq (\pi_0, \pi_1) = \frac{1}{p+q}(q, p)$. The higher the flipping probability $q$, the higher the likelihood for the chain to be in the state 0 at the steady state. Similarly for the state 1. We can verify that $\boldsymbol{\pi}$ indeed satisfies $\boldsymbol{\pi}\boldsymbol{P} = \boldsymbol{\pi}$. For brevity, we drop the dependence on $(p, q)$ and simply write $\boldsymbol{\pi}$ and $\boldsymbol{P}$ when the parameters are clear from context.

## 3 Framework: Transformers via Markov chains

We present our mathematical framework for a principled analysis of transformers via Markov chains. In this paper we focus on first-order binary Markovian data and single-layer transformers though our framework readily generalizes to higher orders and deeper architectures (Sec. 4.4), and multi-state Markov chains (Sec. 4.3).

**Data.** We assume that the vocabulary $\mathcal{X} = \{0, 1\}$ and the input data $\{x_n\}_{n=1}^N \sim (\boldsymbol{\pi}(p, q), \boldsymbol{P}(p, q))$, for some fixed sequence length $N \geq 1$ and $(p, q) \in (0, 1)^2$. Recall that $p + q$ is the switching factor. The parameters $p$ and $q$ provide a tractable mechanism to control the input data, which plays a crucial role in transformer learning.

**Model.** We consider a single-layer transformer with a single-head attention, without layer norm. As the input is binary, the Embedding layer can be simplified to

$$\boldsymbol{x}_n = x_n \, \boldsymbol{e}_1 + (1 - x_n) \, \boldsymbol{e}_0 + \widetilde{\boldsymbol{p}}_n = x_n \, \boldsymbol{e} + \boldsymbol{p}_n, \qquad \text{(Uni-embedding)}$$

where $\boldsymbol{e} \triangleq \boldsymbol{e}_1 - \boldsymbol{e}_0$ is the embedding vector and $\boldsymbol{p}_n \triangleq \boldsymbol{e}_0 + \widetilde{\boldsymbol{p}}_n$ is the new positional encoding. Note that $x_n \in \{0, 1\}$ and hence the embedding is either $\boldsymbol{e} + \boldsymbol{p}_n$ or just $\boldsymbol{p}_n$ depending on $x_n$. The other layers are the same as in Sec. 2.1:

$$x_n \in \{0, 1\} \xrightarrow{\text{Uni-embedding}} \boldsymbol{x}_n \xrightarrow{\text{Attention}} \boldsymbol{y}_n \xrightarrow{\text{FF}} \boldsymbol{z}_n \xrightarrow{\text{Linear}} \text{logit}_n \xrightarrow{\text{Prediction}} f_{\bar{\boldsymbol{\theta}}}(x_1^n). \qquad (2)$$

Let $\bar{\boldsymbol{\theta}} \triangleq (\boldsymbol{e}, \{\boldsymbol{p}_n\}_{n=1}^N, \ldots, b, \boldsymbol{a}) \in \mathbb{R}^D$ denote the joint list of the parameters from all the layers, with $D$ being the total dimensionality. In training large language models, it is a common practice to tie the embedding and linear layer weights, i.e. $\boldsymbol{a} = \boldsymbol{e}$, referred to as *weight tying* (Press and Wolf, 2017). In this case, the list of all parameters, $\boldsymbol{\theta} = (\boldsymbol{e} = \boldsymbol{a}, \{\boldsymbol{p}_n\}_{n=1}^N, \ldots, b) \in \mathbb{R}^{D-d}$, since $\boldsymbol{a}$ is no longer a free parameter. We analyze both weight-tied and general cases.

**Loss.** We consider the cross-entropy loss $L$ from Eq. (1).

**Objective.** Towards understanding the phenomenon in Fig. 1, we utilize the aforementioned framework to study single-layer transformers. In particular, our objective is to address the following question:

> *Can we characterize the loss landscape of singe-layer transformers when the input is Markovian?*

To build intuition about the loss surface, we first examine its global minima and then provide a detailed characterization of the loss landscape, focusing on local optima, in Sec. 4.

## 3.1 SINGLE-LAYER TRANSFORMERS: GLOBAL MINIMA

Since the loss $L$ in Eq. (1) is the cross-entropy loss, it achieves its minimum when the predictive probability matches the Markov kernel (Lemma 1): $f_{\bar{\boldsymbol{\theta}}}(x_1^n) = \mathbb{P}(x_{n+1} = 1 \mid x_n)$. In other words, this occurs when the transformer outputs the correct transition probabilities. This raises a natural question: *can a single-layer transformer exactly represent a first-order Markov chain?* Intuitively speaking, this seems plausible since the transformer, even with access to the full past information $x_1^n$ at each $n \in [N]$, can rely solely on the current symbol $x_n$ (Sec. 2). The following result confirms this intuition, showing that such a realization is indeed a *global minimum* for the loss $L(\cdot)$:

**Theorem 1** (Global minimum). *Let the input sequence be $\{x_n\}_{n=1}^N \sim (\boldsymbol{\pi}(p, q), \boldsymbol{P}(p, q))$ for some fixed $(p, q) \in (0, 1)^2$ and $\boldsymbol{\theta} \in \mathbb{R}^{D-d}$ be the transformer parameters for weight-tied case. Then for all $(p, q)$, there exists a $\boldsymbol{\theta}_\star \in \mathbb{R}^{D-d}$ with an explicit construction such that it is a global minimum for the population loss $L(\cdot)$ in Eq. (1) and its prediction matches the Markov kernel, i.e.*

(i) $L(\boldsymbol{\theta}) \geq L(\boldsymbol{\theta}_\star)$ *for all $\boldsymbol{\theta} \in \mathbb{R}^{D-d}$, and*

(ii) $\mathbb{P}_{\boldsymbol{\theta}_\star}(x_{n+1} = 1 \mid x_1^n) = \mathbb{P}(x_{n+1} = 1 \mid x_n)$, *the Markov kernel or the bigram.*

*Further, $\boldsymbol{\theta}_\star$ satisfies:*

(iii) $L(\boldsymbol{\theta}_\star) = H(x_{n+1} \mid x_n)$, *the entropy rate of the Markov chain.*

(iv) $\nabla L(\boldsymbol{\theta}_\star) = 0$, *i.e. $\boldsymbol{\theta}_\star$ is a stationary point.*

*In addition, the same result holds for the non-weight-tied case when the parameters are in $\mathbb{R}^D$.*

**Remark 1.** In fact, there exist many such global minima as highlighted in the proof (§ B).

*Proof sketch.* The key idea here is to show that any $\boldsymbol{\theta}$ satisfying $f_{\boldsymbol{\theta}}(x_1^n) = \mathbb{P}(x_{n+1} = 1 \mid x_n)$ is a global minimum and is a stationary point with the loss being the entropy rate (Lemmas. 1 and 2). To construct such a $\boldsymbol{\theta}$, we utilize the fact the Markov kernel is only a function of $x_n$ and thus we can ignore the past information in the Attention layer using only the skip. We defer the full proof to § B. $\qquad \square$

*Empirical evidence for learning the Markov kernel.* As demonstrated in the proof above, a canonical way to realize the Markov kernel by the single-layer transformer is to rely only on the current

symbol $x_n$ and ignore the past in the Attention layer. We now empirically confirm this fact. For our experiments, we use the single-layer transformer (Table 1) and report the results averaged across 5 runs and corresponding to the best set of hyper-parameters after a grid search (Table 2). In particular, for $p = 0.2, q = 0.3$, and $d = 4$, we generate sequences $\{x_n\}_{n=1}^N \sim (\boldsymbol{\pi}(p,q), \boldsymbol{P}(p,q))$ of length $N = 1024$ and train the transformer parameters $\boldsymbol{\theta}$ (weight-tied) to minimize the cross-entropy loss in Eq. (1). At inference, we interpret the attention layer and observe that the relative magnitude of the attention contribution to the final attention output $\boldsymbol{y}_n$ is negligible, i.e. the ratio $\|\boldsymbol{W}_O \sum_{i \in [n]} \text{att}_{n,i} \cdot \boldsymbol{W}_V \boldsymbol{x}_i\| / \|\boldsymbol{y}_n\| \approx 0.01$. Hence, the attention contribution can be neglected compared to the skip-connection, i.e. $\boldsymbol{y}_n \approx \boldsymbol{x}_n$. Using this approximation, in § D we derive a formula for the final predicted probability $f_{\boldsymbol{\theta}}(x_1^n)$ as it is learnt by the network. This formula reveals interesting insights about the learnt parameters of the transformer:

- **Constant embeddings.** The positional embedding $\boldsymbol{p}_n$ is constant across $n$, i.e. it is independent of the sequence position, reflecting the fact that it has learnt to capture the homogeneity of the Markovian chain just from the data.

- **Low-rank weights.** The weight matrices are all approximately rank-one; while it is not fully clear why the training algorithm converges to low rank solutions, they do indeed provide a canonical and simple way to realize the Markov kernel, as illustrated in § D.

Further, we show in § D that plugging in the numerical values obtained from the average of five runs, the probability given by our formula matches the theory, i.e. the model learns to correctly output the Markov kernel probabilities. Indeed, Fig. 3b shows that the test loss of the model converges to the theoretical global minimum (Thm. 1), the entropy rate of the source, corresponding to the bigram, when $p = 0.2$ and $q = 0.3$ ($p + q < 1$). We likewise observe a similar phenomenon without weight tying. For the prediction probability, we focus on the zero positions $n = n_k$ such that $x_{n_k} = 0$. Fig. 3d shows that irrespective of the index $k$ and the past $x_1^{n_k-1}$, if the current bit $x_{n_k}$ is 0, the model correctly predicts the probability for the next bit $x_{n_k+1}$ being 1, which equals $p$ theoretically (Fig. 2b). More precisely, $f_{\boldsymbol{\theta}}(x_1^{n_k-1}, x_{n_k} = 0) = p$ for all $x_1^{n_k-1}$ and $k$, in line with property (ii) of Thm. 1. A similar conclusion holds with $x_{n_k} = 1$ and prediction probability $q$. This indicates that the model has learned to recognize the data as first-order Markovian, relying solely on $x_n$ to predict $x_{n+1}$.

While Thm. 1 and above empirical results highlight the presence of global minima on the loss surface, they does not address local optima, as empirically shown in Fig. 1. We precisely address this in the next section and analyze the loss landscape in terms of the local optima.

## 4 SINGLE-LAYER TRANSFORMERS: LOCAL OPTIMA

In this section we present our main results about the loss landscape of single-layer transformers in terms of local optima. In particular, we prove the existence of bad local minima and saddle points on the loss surface (Thms. 2 and 3), in addition to the global minima discussed above (Thm. 1). Interestingly, the presence of these local optima is influenced by two key factors: *switching factor* of the Markov chain and the *weight tying* of the transformer, highlighting the intricate interplay between the input data and the model architecture. First, we present the results for the weight tying scenario.

### 4.1 WEIGHT TYING: BAD LOCAL MINIMA

When the embedding and linear layers are tied, i.e. $\boldsymbol{e} = \boldsymbol{a}$, our analysis reveals the following surprising fact: if the switching factor $p + q$ is greater than one, there exist *bad local minima* $\boldsymbol{\theta}_{\boldsymbol{\pi}} \in \mathbb{R}^{D-d}$, where the prediction probability $f_{\boldsymbol{\theta}_{\boldsymbol{\pi}}}(\cdot)$ is the marginal stationary distribution $\boldsymbol{\pi}$ (unigram), disregarding the past and the present information (Thm. 2 and Fig. 3c). Now we state the result formally. Let $L_\star \triangleq L(\boldsymbol{\theta}_\star)$ denote the global minimal loss from Thm. 1.

**Theorem 2** (Bad local minimum). *Let the input sequence be $\{x_n\}_{n=1}^N \sim (\boldsymbol{\pi}(p,q), \boldsymbol{P}(p,q))$ for a fixed $(p,q) \in (0,1)^2$ and the transformer parameters be weight-tied. If $p + q > 1$, there exists an explicit $\boldsymbol{\theta}_{\boldsymbol{\pi}} \in \mathbb{R}^{D-d}$ such that it is a bad local minimum for the loss $L(\cdot)$, i.e.*

(i) *there exists a neighborhood $\mathcal{B}(\boldsymbol{\theta}_{\boldsymbol{\pi}}, r)$ with $r > 0$ such that $L(\boldsymbol{\theta}) \geq L(\boldsymbol{\theta}_{\boldsymbol{\pi}})$ for all $\boldsymbol{\theta} \in \mathcal{B}(\boldsymbol{\theta}_{\boldsymbol{\pi}}, r)$, with $L(\boldsymbol{\theta}_{\boldsymbol{\pi}}) > L_\star$.*

*Further, $\boldsymbol{\theta}_{\boldsymbol{\pi}}$ satisfies:*

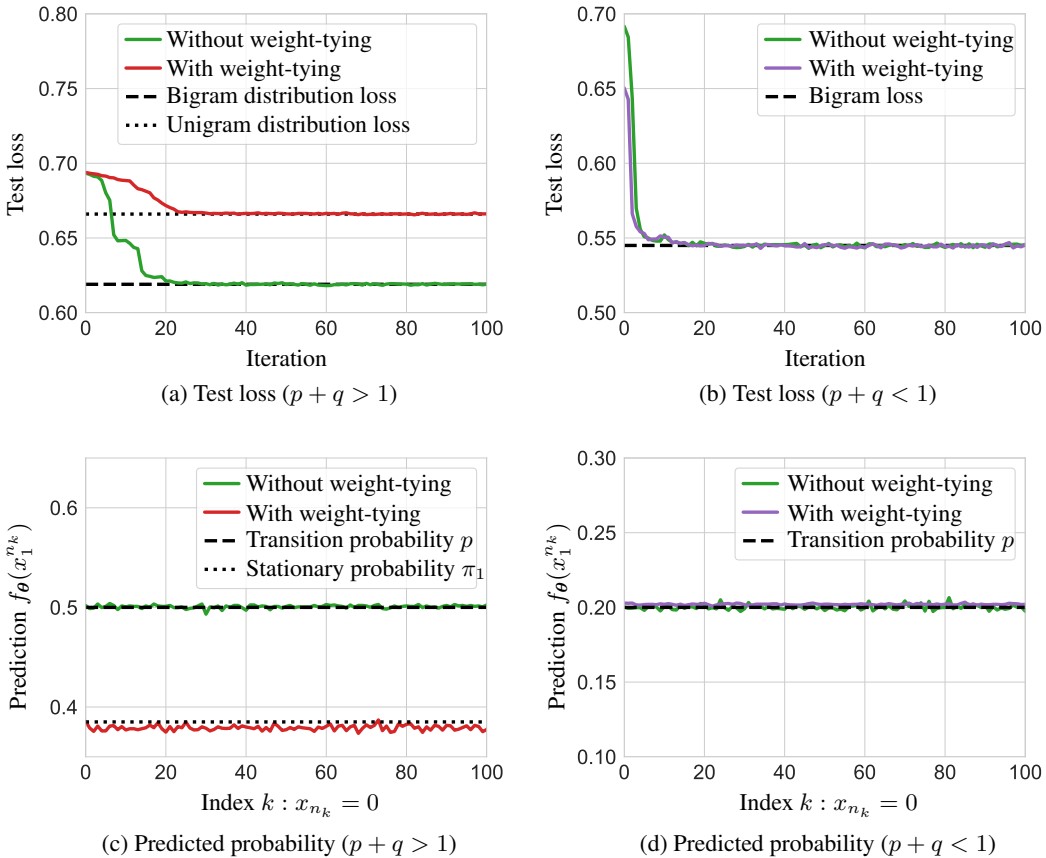

Figure 3: Effect of weight tying on test loss and predicted probabilities $f_{\boldsymbol{\theta}}(x_1^{n_k})$ for zero indices $\{n_k\}_{k=1}^{100}$ such that $x_{n_k} = 0$. For (a),(c): $p = 0.5, q = 0.8$. With weight tying, the loss converges to a local minimum, and the predicted probability is $\pi_1 = p/(p+q)$. Without weight tying, we predict the correct probability $p$ and converge to a global minimum. For (b),(d): $p = 0.2, q = 0.3$. The test loss always converges to a global minimum, and the predicted probability is $p$.

(ii) $\mathbb{P}_{\boldsymbol{\theta}_{\boldsymbol{\pi}}}(x_{n+1} = 1 \mid x_1^n) = \mathbb{P}(x_{n+1} = 1) = \pi_1$, *the marginal distribution or the unigram.*

(iii) $L(\boldsymbol{\theta}_{\boldsymbol{\pi}}) = H(x_{n+1}) = H(\boldsymbol{\pi})$, *the entropy of the marginal.*

(iv) $\nabla L(\boldsymbol{\theta}_{\boldsymbol{\pi}}) = 0$, *i.e. $\boldsymbol{\theta}_{\boldsymbol{\pi}}$ is a stationary point.*

**Remark 2.** Since $L(\boldsymbol{\theta}_{\boldsymbol{\pi}}) = H(x_{n+1})$ and $L_\star = H(x_{n+1}|x_n)$, the optimality gap $L(\boldsymbol{\theta}_{\boldsymbol{\pi}}) - L_\star = H(x_{n+1}) - H(x_{n+1}|x_n) = I(x_n; x_{n+1}) \geq 0$, where $I(x_n; x_{n+1})$ is the mutual information between $x_n$ and $x_{n+1}$ (Cover and Thomas, 2006). It equals zero if and only if $x_n$ and $x_{n+1}$ are independent, which happens for $p + q = 1$ (since $\mathbb{P}(x_{n+1} = 1 \mid x_n) = x_n(1 - p - q) + p$).

*Proof sketch.* The main idea behind constructing $\boldsymbol{\theta}_{\boldsymbol{\pi}}$ is that if we set $\boldsymbol{e} = \boldsymbol{a} = 0$ in the Linear layer, the model ignores the inputs all together and outputs a constant probability, and in particular $\pi_1$ by choosing the bias $b$ appropriately, i.e. $f_{\boldsymbol{\theta}_{\boldsymbol{\pi}}}(x_1^n) = \pi_1$ for all $x_1^n, n$. For this $\boldsymbol{\theta}_{\boldsymbol{\pi}}$ it's easy to show that $L(\boldsymbol{\theta}_{\boldsymbol{\pi}}) = H(\boldsymbol{\pi})$ and that it's a stationary point. Further we show that the Hessian at $\boldsymbol{\theta}_{\boldsymbol{\pi}}$ follows the structure $\begin{bmatrix} \boldsymbol{H}_{\boldsymbol{\alpha}} & 0 \\ 0 & 0 \end{bmatrix}$ where $\boldsymbol{H}_{\boldsymbol{\alpha}} \succ 0$ when $p + q > 1$, and that it implies the local minimality of $\boldsymbol{\theta}_{\boldsymbol{\pi}}$. We defer the full proof to § B.4. □

*Empirical evidence for bad local minima.* The proof of Thm. 2 above highlights that when the linear weight $\boldsymbol{a}$ is zero in $\boldsymbol{\theta}$, it serves as a bad local minimum. While this might seem as a theoretical anamoly, we empirically confirm that it is not. Specifically, we use the same weight-tied setting

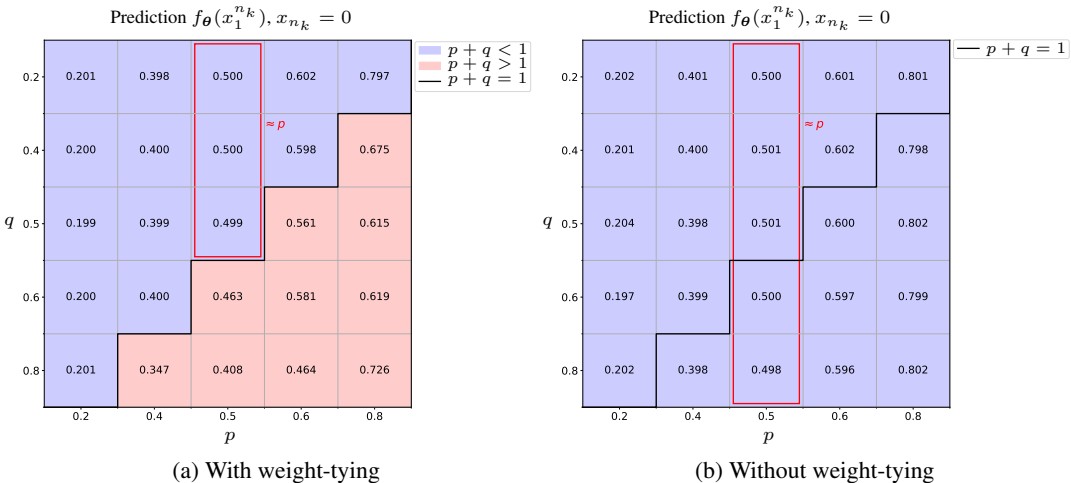

Figure 4: Average of predicted probabilities across 5 runs for different values of $p$ and $q$, with and without weight-tying. In the former case, there is a clear demarcation beteen the cases where $p + q < 1$ and those where $p + q > 1$. For $p + q < 1$, all runs accurately predict the correct conditional probability. For $p + q > 1$, some of the runs predict the stationary probability instead, causing the average to diverge from the correct $p$. In the latter case, the model always predicts the correct probability for all $p$ and $q$.

as before but with the flipping probabilities $p = 0.5$ and $q = 0.8$ instead, we observe that the magnitude of the vector $\boldsymbol{a}$ is approximately $0.01$ whereas that of $\boldsymbol{z}_n$ is $4.0$ in the Linear layer. Thus, $\langle \boldsymbol{a}, \boldsymbol{z}_n \rangle \approx 0.04$ , while the bias term $b \approx -0.4$ implying $\sigma(\langle \boldsymbol{a}, \boldsymbol{z}_n \rangle + b) \approx \sigma(b)$. Hence the final prediction returned by the network only depends on the bias of the linear layer, totally independent of the input sequence $x_1^N$. Fig. 3c illustrates this fact and shows that the model always predicts the stationary probability $\pi_1$ at all positions in the sequence, independent of the input. Here we plot for the indices $k$ such that $x_{n_k} = 0$ but we observe the same phenomenon for $x_{n_k} = 1$, i.e. $f_{\boldsymbol{\theta}}(x_1^n) = \pi_1$ for all $x_1^n, n$, thus verifying property (ii) of Thm. 2. Similarly, Fig. 3a demonstrates that the model test loss converges to the entropy of the stationary distribution $H(\boldsymbol{\pi})$, the unigram loss, instead of the global minimum $L_\star$ given by the entropy rate of the source (Thm. 2 - (iii) ). Fig. 4 illustrates for a wider range of $(p, q) \in (0, 1)^2$.

*Interpreting global and local minima.* Thm. 2 should be interpreted in the light that it guarantees the existence of bad local minima only for $p + q > 1$ (in sync with the experiments, Fig. 4 ). While there could exist such minima even when $p + q < 1$, we empirically observe that the model always converges to the global minimum in this scenario (Fig. 3b). Indeed, Makkuva et al. (2024) shows that local minima also exist for $p + q < 1$, however, the standard Gaussian initialization around zero falls outside their basin of attraction, and hence gradient based methods escape them. Likewise, while Thm. 1 guarantees the existence of global minima for all $(p, q)$ and in particular for $p + q > 1$, empirically the model often converges to bad local minima as highlighted above (Fig. 3a).

## 4.2 WITHOUT WEIGHT TYING: SADDLE POINTS

Now we let the token-embedding $\boldsymbol{e} \in \mathbb{R}^d$ and the linear weight $\boldsymbol{a} \in \mathbb{R}^d$ be independent parameters. Interestingly, in this scenario, the earlier local minimum $\boldsymbol{\theta}_{\boldsymbol{\pi}}$ becomes a saddle point.

**Theorem 3** (Saddle point). *Consider the same setting as in Thm. 2 and for $p + q > 1$, let $\boldsymbol{\theta}_{\boldsymbol{\pi}} = (\boldsymbol{e}_{\boldsymbol{\pi}} = \boldsymbol{a}_{\boldsymbol{\pi}}, \ldots, b_{\boldsymbol{\pi}}) \in \mathbb{R}^{D-d}$ be the corresponding bad local minimum for the loss $L(\cdot)$ in the weight-tied scenario. Then its extension $\bar{\boldsymbol{\theta}}_{\boldsymbol{\pi}} \triangleq (\boldsymbol{\theta}_{\boldsymbol{\pi}}, \boldsymbol{a}_{\boldsymbol{\pi}}) \in \mathbb{R}^D$ is a saddle point for $L(\cdot)$ in $\mathbb{R}^D$ in the non-weight-tied case. Further, $\bar{\boldsymbol{\theta}}_{\boldsymbol{\pi}}$ satisfies the same properties (ii)–(iv) as in Thm. 2.*

*Empirical evidence and interpretation.* In view of the theoretical results above, removing weight tying is possibly beneficial: the bad local minimum in the weight-tied case for $p + q > 1$ suddenly becomes a saddle point when the weight tying is removed. We observe a similar phenomenon empirically (Fig. 4): as shown in Fig. 3a, when not weight-tied, the model's test loss converges to

the entropy rate of the source when $p + q > 1$, in contrast to the weight-tied case, possibly escaping the saddle point (Thm. 3). The fact that the model eventually learns the correct Markovian kernel is further demonstrated by the red curve in Fig. 3c. Figs. 3b and 3a together highlight that the model always (empirically) converges to the global minimum in the non-weight-tied case irrespective of the switching factor $p + q$.

**Key insights.** Together, our theoretical and empirical results highlight that when the switching $p + q > 1$, the weight-tied model can get stuck at bad local minima corresponding to the unigram. In contrast, the non-weight-tied model can potentially escape saddle points to converge to the global minima, corresponding to the bigram (Markov kernel). This explains the phenomenon in Fig. 1 for the single-layer transformer, where $p = 0.5$ and $q = 0.8$. When $p + q < 1$, we empirically observe that the model always converges to a global minimum irrespective of weight-tying.

## 4.3 MULTI-STATE MARKOV CHAINS

Our framework focusing on the binary setting $\mathcal{X} = \{0, 1\}$ above, readily generalizes to the multi-state setting $\mathcal{X} = \{0, 1, \ldots, S - 1\}$, where $S$ is the state (vocabulary) size. Let the input $(x_n)_{n \geq 1} \sim \boldsymbol{P}(p)$ be a first-order Markov chain, where either $x_{n+1} = x_n$ with probability $1 - p$, or $x_{n+1}$ switches to a different state in the vocabulary uniformly randomly with probability $p/(S - 1)$, for some $p \in (0, 1)$. The symmetric kernel $\boldsymbol{P}(p)$ generalizes that of the binary Markovian case, $\boldsymbol{P}(p, q)$ in Sec. 2.2, when $p = q$. Fig. 5 illustrates this for the five-state setting $S = 5$. For the transformer, we use the same architecture as in Sec. 2.1 except for the fact that we now have $S$ token embeddings in the embedding and linear layers.

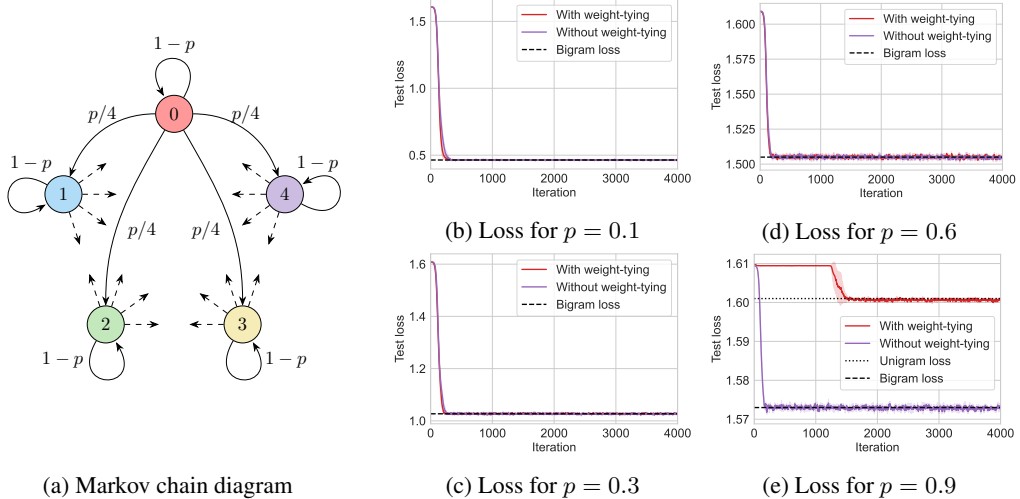

(a) Markov chain diagram    (b) Loss for $p = 0.1$    (d) Loss for $p = 0.6$

(c) Loss for $p = 0.3$    (e) Loss for $p = 0.9$

Figure 5: Effect of weight-tying for the multi-state Markov chain with $S = 5$ states. (a) shows the symmetric multi-state Markov data model used, while (b), (c), (d) and (e) show the test loss for different values of $p$. Similar to the binary case, when $p$ is large enough, a 1-layer transformer with weight-tying gets stuck in local minima, whereas it escapes to global minima without.

**Results.** When a single-layer transformer is trained on a multi-state Markov chain with $S = 5$ for various values of $p \in \{0.1, 0.3, 0.6, 0.9\}$, Fig. 5 (analogous to Fig. 3) illustrates that models with weight tying tend to get stuck in local minima (unigram), whereas those without converge to global minima (bigram). As discussed in Sec. 4.1 for binary case, the switching condition $p + q > 1$ is equivalent to $p > 0.5$ when $p = q$, where the model empirically gets trapped in local minima. Interestingly, for the five-state case here, we observe convergence to global minima even at $p = 0.6 > 0.5$, suggesting that the switching threshold exceeds 0.5 in this scenario. In the binary case, this threshold of 0.5 was derived theoretically via Hessian computation (Eq. (13)). In a similar vein, the proofs of Thm. 2 and Thm. 3 which extend to the multi-state setting, yield a comparable condition via Hessian analysis. Together, these findings highlight that similar theoretical results could be established highlighting the role of weight tying for multi-state scenarios as well, in congruence with

above empirical findings. Obtaining an explicit mathematical expression for the switching threshold, however, is an exciting direction for future research and beyond the current scope of this paper.

### 4.4 Does depth help escape local minima?

For single-layer transformers, the aforementioned results highlight the significance of the switching factor and the weight tying on the loss landscape. In stark contrast, we empirically observe that for transformers of depth 2 and beyond, the loss curve eventually reaches the global minimum regardless of these factors, as highlighted in Fig. 1. Interestingly, we observe during training that it first reaches a plateau at a loss value corresponding to $H(\pi)$ and after a few additional iterations, it further drops down until it reaches the global minimum (the entropy rate). This suggests that, while there could still be local minima, increasing the number of layers positively affects the loss curvature at the bad local minima in a manner making it easier to escape and reach the global minimum. In the context of feed-forward neural networks, depth of the architecture has been shown to play a major role in terms of the representation power and learning capabilities (Telgarsky, 2016; Arora et al., 2019). Given our empirical observations, a similar analysis for transformers that demonstrates the benefits of depth is an intriguing direction for future research.

## 5 Related work

There is tremendous interest and active research in understanding transformer models from various perspectives (Weiss et al., 2021; Giannou et al., 2023; Oymak et al., 2023; Li et al., 2023a; Fu et al., 2023; Noci et al., 2023; Tian et al., 2023). Yun et al. (2020); Pérez et al. (2021); Wei et al. (2022); Malach (2023); Jiang and Li (2023) demonstrate the representation capabilities of transformers and show properties such as universal approximation and Turing-completeness. Another line of inquiry (Elhage et al., 2021; Snell et al., 2021; Wang et al., 2023; Geva et al., 2023) is mechanistic interpretability, i.e. reverse-engineering transformer operations on specific synthetic tasks (e.g., matrix inversion and eigenvalue decomposition in Charton (2022), modular addition in Nanda et al. (2023)) to understand the transformer components but they usually lack theoretical guarantees, as opposed to ours. Li et al. (2023c) studies how transformers learn semantic structures across words while we are interested in how they learn sequentiality in input data. Tarzanagh et al. (2023a;b) take an optimization-theoretic perspective to study training dynamics and characterize implicit biases in transformer models trained with gradient descent. In contrast, we characterize the local and global minima of the loss landscape of these models under sequential input data.

On the other hand, Chen et al. (2024); Bietti et al. (2023); Dong et al. (2023); Akyürek et al. (2023); Von Oswald et al. (2023); Xie et al. (2021); Bai et al. (2023); Li et al. (2023b); Garg et al. (2022) study in-context learning capabilities of transformers. Along this line, our work is closely related to Nichani et al. (2024); Bietti et al. (2023); Edelman et al. (2024) in the Markovian setup, however, while they focus on in-context learning, which is apparent for two or more layers, we investigate the loss landscape of single-layer models. Chung et al. (2021) provide empirical evidence to suggest that weight tying has drawbacks in encoder-only models, which is in line with our observations that removing weight tying is beneficial in decoder-only models with Markovian input data. More recently, Rajaraman et al. (2024b) study the effect of tokenization on learning Markov chains. Ildiz et al. (2024) show an equivalence between the attention mechanism and Markov models, whereas we characterize the loss landscape of attention-based transformers when the input is Markovian.

## 6 Conclusion and Open questions

In this work, we provide a novel framework for a systematic theoretical and empirical study of the sequential modeling capabilities of transformers through Markov chains. Leveraging this framework, we theoretically characterize the loss landscape of single-layer transformers and show the existence of global minima and bad local minima contingent upon the specific data characteristics and the transformer architecture, and independently verify them by experiments. We further reveal interesting insights for deeper architectures. We believe our framework provides a new avenue for a principled study of transformers. In particular, some interesting open questions in this direction include establishing similar results for higher-order Markov chains and deeper transformer models, showcasing the interplay between the model depth and Markovian order.

ACKNOWLEDGEMENTS

The work was supported in part by the Swiss National Science Foundation under Grant 200364.

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

**Organization.** The appendix is organized as follows:

- App. A details the transformer architecture, especially that of the attention mechanism.
- App. B provides the proofs for our theoretical results on first-order Markov chains.
- App. C contains additional experimental details and results for first-order Markov chains.

## A  THE TRANSFORMER ARCHITECTURE

We describe the Transformer architecture from Sec. 2.1 in detail, using the embedding layer simplication from Sec. 3:

$$\boldsymbol{x}_n = x_n \, \boldsymbol{e} + \boldsymbol{p}_n \in \mathbb{R}^d, \qquad \text{(Uni-embedding)}$$

$$\boldsymbol{y}_n = \boldsymbol{x}_n + \boldsymbol{W}_O \sum_{i \in [n]} \text{att}_{n,i} \cdot \boldsymbol{W}_V \, \boldsymbol{x}_i \in \mathbb{R}^d, \qquad \text{(Attention)}$$

$$\boldsymbol{z}_n = \boldsymbol{y}_n + \boldsymbol{W}_2 \, \text{ReLU}(\boldsymbol{W}_1 \, \boldsymbol{y}_n) \in \mathbb{R}^d, \qquad \text{(FF)}$$

$$\text{logit}_n = \langle \boldsymbol{a}, \boldsymbol{z}_n \rangle + b \qquad\qquad\;\; \in \mathbb{R}, \qquad \text{(Linear)}$$

$$f_{\bar{\boldsymbol{\theta}}}(x_1^n) \triangleq \mathbb{P}_{\bar{\boldsymbol{\theta}}}\left(x_{n+1} = 1 \mid x_1^n\right) = \underbrace{\sigma(\text{logit}_n)}_{\in [0,1]}. \qquad \text{(Prediction)}$$

**(i) Embedding:** The discrete tokens $x_n = 1$ and $x_n = 0$ are mapped to the token-embeddings $\boldsymbol{e}$ and $0$ in $\mathbb{R}^d$ respectively, where $d$ is the embedding dimension. The positional embedding $\boldsymbol{p}_n \in \mathbb{R}^d$ encodes the positional information (varies with $n$). The sum of these two embeddings constitutes the final input embedding $\boldsymbol{x}_n \in \mathbb{R}^d$.

**(ii) Attention:** The attention layer can be viewed as mappping a query and a set of key-value pairs to an output, which are all vectors Vaswani et al. (2017). That is, on top of the skip-connection $\boldsymbol{x}_n$, the output $\boldsymbol{y}_n \in \mathbb{R}^d$ is computed as a weighted sum of the values $\boldsymbol{v}_i \triangleq \boldsymbol{W}_V \, \boldsymbol{x}_i$. The weight assigned to each value, $\text{att}_{n,i}$, is computed by a compatibility function of the query vector $\boldsymbol{q}_n \triangleq \boldsymbol{W}_Q \, \boldsymbol{x}_n$ and the corresponding key vectors $\boldsymbol{k}_i \triangleq \boldsymbol{W}_K \, \boldsymbol{x}_i$ for all $i \in [n]$. More precisely, $\text{att}_{n,i} \triangleq \text{softmax}((\langle \boldsymbol{q}_n, \boldsymbol{k}_1 \rangle, \dots, \langle \boldsymbol{q}_n, \boldsymbol{k}_n \rangle)/\sqrt{d})_i$. $\boldsymbol{W}_{K,Q,V} \in \mathbb{R}^{m \times d}$ are the respective key, query, and value matrices, and $\boldsymbol{W}_O \in \mathbb{R}^{d \times m}$ is the projection matrix. For multi-headed attention, the same operation is performed on multiple parallel heads, whose outputs are additively combined.

**(iii) Feed-forward (FF):** The FF transformation consists of a skip-connection and a single-hidden layer with ReLU activation and weight matrices $\boldsymbol{W}_2 \in \mathbb{R}^{d \times r}$, and $\boldsymbol{W}_1 \in \mathbb{R}^{r \times d}$. The FF layer is applied token-wise on each $\boldsymbol{y}_n \in \mathbb{R}^d$ to output $\boldsymbol{z}_n \in \mathbb{R}^d$ with the same dimensionality.

**(iv) Linear:** The linear layer transforms the final output embedding $\boldsymbol{z}_n$ to a scalar $\text{logit}_n \in \mathbb{R}$, with the weight parameter $\boldsymbol{a} \in \mathbb{R}^d$ and the bias $b \in \mathbb{R}$.

**(v) Prediction:** The sigmoid activation finally converts the scalar logits to probabilities for the next-token prediction. Since the vocabulary has only two symbols, it suffices to compute the probability for the symbol 1: $f_{\bar{\boldsymbol{\theta}}}(x_1^n) \triangleq \mathbb{P}_{\bar{\boldsymbol{\theta}}}(x_{n+1} = 1 \mid x_1^n) = \sigma(\text{logit}_n) \in [0,1]$. More generally, the logits are of the same dimensionality as the vocabulary and are converted to the prediction probabilities using a softmax layer, which simplifies to the sigmoid for the binary case. Likewise, there are as many token-embeddings as the words in the vocabulary and several layers of multi-headed attention and FF operations are applied alternatively on the input embeddings to compute the final logits.

Finally, The Transformer parameters $\bar{\boldsymbol{\theta}} \triangleq (\boldsymbol{e}, \{\boldsymbol{p}_n\}_{n=1}^N, \dots, b, \boldsymbol{a}) \in \mathbb{R}^D$ are trained via the cross-entropy loss on the next-token prediction:

$$L(\bar{\boldsymbol{\theta}}) \triangleq -\frac{1}{N} \sum_{n \in [N]} \mathbb{E}_{x_1^{n+1}}[x_{n+1} \cdot \log f_{\bar{\boldsymbol{\theta}}}(x_1^n) + (1 - x_{n+1}) \cdot \log(1 - f_{\bar{\boldsymbol{\theta}}}(x_1^n))], \qquad (3)$$

# B  PROOFS OF SEC. 4

We now present our proofs for the technical results in Sec. 4. Towards this, we first establish two useful lemmas on the loss function $L(\cdot)$ and the corresponding gradient computation. Let $\bar{\boldsymbol{\theta}} = (\boldsymbol{e}, \{\boldsymbol{p}_n\}_{n=1}^N, \ldots, b, \boldsymbol{a}) \in \mathbb{R}^D$ be the list of parameters in the non-weight-tied case and $\boldsymbol{\theta} = (\boldsymbol{e} = \boldsymbol{a}, \{\boldsymbol{p}_n\}_{n=1}^N, \ldots, b) \in \mathbb{R}^{D-d}$ in the weight-tied case. With a slight abuse of notation, by $\boldsymbol{w} \in \bar{\boldsymbol{\theta}}$ we mean a specific parameter $\boldsymbol{w}$ among the set $\{\boldsymbol{e}, \boldsymbol{p}_1, \ldots, \boldsymbol{p}_N, \ldots, b, \boldsymbol{a}\}$. Since the weight-tied scenario is a special case of the non-weight-tied one with $\boldsymbol{e} = \boldsymbol{a}$, we directly present the results for the general non-weight-tied case, but both lemmas hold for $\boldsymbol{\theta} \in \mathbb{R}^{D-d}$ as well. First we start with the loss function.

**Lemma 1** (**Loss as KL divergence**). *Let the input sequence be $\{x_n\}_{n=1}^N \sim (\boldsymbol{\pi}(p,q), \boldsymbol{P}(p,q))$ for some fixed $(p,q) \in (0,1)^2$, $\bar{\boldsymbol{\theta}} = (\boldsymbol{e}, \{\boldsymbol{p}_n\}_{n=1}^N, \ldots, b, \boldsymbol{a}) \in \mathbb{R}^D$ be the full list of the transformer parameters, and $L(\bar{\boldsymbol{\theta}})$ be the corresponding cross-entropy loss in Eq. (1). Then the loss function $L(\cdot)$ is equivalent to the KL divergence between the Markov kernel and the predicted distribution, i.e.*

$$L(\bar{\boldsymbol{\theta}}) = \frac{1}{N} \sum_{n \in [N]} \mathbb{E}_{x_1^n} \left[ D_{\mathrm{KL}}(\mathbb{P}(x_{n+1} = \cdot \mid x_n) \parallel \mathbb{P}_{\bar{\boldsymbol{\theta}}}(x_{n+1} = \cdot \mid x_1^n)) \right] + H(x_{n+1} | x_n), \quad (4)$$

*where $D_{\mathrm{KL}}(P \parallel Q)$ is the KL divergence between two distributions $P$ and $Q$, and $H(x_{n+1}|x_n)$ is the entropy rate of the Markov chain.*

**Remark 3.** Consequently, Eq. (4) highlights that any parameter $\bar{\boldsymbol{\theta}}$ with the predicted probability $f_{\bar{\boldsymbol{\theta}}}(x_1^n) = \mathbb{P}(x_{n+1} = 1 \mid x_n)$ is a global minimum for the loss $L$ as $D_{\mathrm{KL}}(\cdot \parallel \cdot) \geq 0$ Cover and Thomas (2006). We utilize this fact in the proof of Thm. 1 below.

*Proof.* We defer the proof to § B.6. $\qquad\square$

**Lemma 2** (**Gradient computation**). *Consider the same data and parameter setting as in Lemma 1 and $L(\bar{\boldsymbol{\theta}})$ be the cross-entropy loss in Eq. (1). Then for any parameter $\boldsymbol{w} \in \bar{\boldsymbol{\theta}}$,*

$$\begin{aligned}
\nabla_{\boldsymbol{w}} L(\bar{\boldsymbol{\theta}}) &= -\frac{1}{N} \sum_{n \in [N]} \mathbb{E}_{x_1^{n+1}} \left[ (x_{n+1} - f_{\bar{\boldsymbol{\theta}}}(x_1^n)) \cdot \nabla_{\boldsymbol{w}} \left( \boldsymbol{a}^\top \boldsymbol{z}_n + b \right) \right] \\
&= -\frac{1}{N} \sum_{n \in [N]} \mathbb{E}_{x_1^n} \left[ (\mathbb{P}(x_{n+1} = 1 \mid x_n) - f_{\bar{\boldsymbol{\theta}}}(x_1^n)) \cdot \nabla_{\boldsymbol{w}} \left( \boldsymbol{a}^\top \boldsymbol{z}_n + b \right) \right].
\end{aligned} \quad (5)$$

**Remark 4.** Eq. (5) highlights that any parameter $\bar{\boldsymbol{\theta}}$ with the predicted probability $f_{\bar{\boldsymbol{\theta}}}(x_1^n) = \mathbb{P}(x_{n+1} = 1 \mid x_n)$ is also a stationary point for the loss $L$. We utilize this fact in the proof of Thm. 1 below.

*Proof.* We defer the proof to § B.7. $\qquad\square$

We now detail the proofs of theorems in Sec. 4. We prove the global minimum result in Thm. 1 in two parts, separately for the cases when $p + q \leq 1$ and $p + q > 1$. For both these cases, first we consider weight-tying and in App. B.3, the non-tied case.

## B.1  PROOF OF THM. 1 FOR $p + q \leq 1$, WEIGHT-TYING

*Proof.* We assume that $p + q \leq 1$ and that we use weight tying, i.e. the list of parameters $\boldsymbol{\theta} = (\boldsymbol{e} = \boldsymbol{a}, \{\boldsymbol{p}_n\}_{n=1}^N, \ldots, b) \in \mathbb{R}^{D-d}$. Thus in view of Lemma 1 and Lemma 2, it follows that any $\boldsymbol{\theta}$ satisfying $f_{\boldsymbol{\theta}}(x_1^n) = \mathbb{P}(x_{n+1} = 1 \mid x_n)$ is a global minimum with loss equalling the entropy rate, and is a stationary point. Hence it suffices to construct such a $\boldsymbol{\theta}$.

To build our intuition towards designing $\boldsymbol{\theta}_\star$, recall that the Markov kernel $\mathbb{P}(x_{n+1} = 1 \mid x_n)$ can be succinctly written as $\mathbb{P}(x_{n+1} = 1 \mid x_n) = x_n(1-p-q)+p$. To ensure that $f_{\boldsymbol{\theta}}(x_1^n) = x_n(1-p-q)+p$, it suffices for the transformer to utilize only the information from the current symbol $x_n$ and ignore the past $x_1^{n-1}$. In view of the transformer architecture in § A, a natural way to realize this is to let $\boldsymbol{W}_O = 0$ and $\boldsymbol{W}_2 = 0$ in Attention and FF respectively. This implies that $\boldsymbol{z}_n = \boldsymbol{y}_n = \boldsymbol{x}_n = x_n \boldsymbol{e} + \boldsymbol{p}_n$. Hence

the logits are given by $\mathrm{logit}_n = \langle \boldsymbol{e}, \boldsymbol{z}_n \rangle + b = x_n \|\boldsymbol{e}\|^2 + \langle \boldsymbol{e}, \boldsymbol{p}_n \rangle + b$. Since $f_{\boldsymbol{\theta}}(x_1^n) = \sigma(\mathrm{logit}_n)$ and it equals the Markov kernel, we have that

$$\sigma(\mathrm{logit}_n) = \sigma(x_n \|\boldsymbol{e}\|^2 + \langle \boldsymbol{e}, \boldsymbol{p}_n \rangle + b) = x_n(1 - p - q) + p.$$

Rewriting,

$$x_n \|\boldsymbol{e}\|^2 + \langle \boldsymbol{e}, \boldsymbol{p}_n \rangle + b = \log \left( \frac{x_n(1 - p - q) + p}{(1 - p)(1 - x_n) + q x_n} \right), \quad x_n \in \{0, 1\}.$$

Substituting $x_n = 0$ and $x_n = 1$, we further simiplify to

$$\langle \boldsymbol{e}, \boldsymbol{p}_n \rangle + b = \log \left( \frac{p}{1 - p} \right),$$

$$\|\boldsymbol{e}\|^2 + \langle \boldsymbol{e}, \boldsymbol{p}_n \rangle + b = \log \left( \frac{1 - q}{q} \right).$$

Subtracting both the equations we obtain that a global minimum $\boldsymbol{\theta}$ should satisfy

$$\|\boldsymbol{e}\|^2 = \log \left( \frac{(1 - p)(1 - q)}{pq} \right),$$

$$\langle \boldsymbol{e}, \boldsymbol{p}_n \rangle + b = \log \left( \frac{p}{1 - p} \right). \tag{6}$$

Note the above choice of $\boldsymbol{e}$ is well-defined since $\frac{(1-p)(1-q)}{pq} > 1$ when $p + q < 1$ and hence $\log \frac{(1-p)(1-q)}{pq} > 0$. While there exist infinitely many solutions for $(\boldsymbol{e}, \boldsymbol{p}_n, b)$ satisfying Eq. (6), a canonical such solution for the global minimum $\boldsymbol{\theta} = \boldsymbol{\theta}_\star$ is

$$\boldsymbol{\theta}_\star = \left( \boldsymbol{e} = \boldsymbol{a} = \mathbf{1} \sqrt{\frac{1}{d} \log \frac{(1 - p)(1 - q)}{pq}}, \{\boldsymbol{p}_n = 0\}_{n=1}^N, \boldsymbol{W}_O = 0, \boldsymbol{W}_{K,Q,V}, \boldsymbol{W}_2 = 0, \boldsymbol{W}_1, b = \log \frac{p}{1 - p} \right), \tag{7}$$

where $\mathbf{1} \in \mathbb{R}^{D-d}$ denotes the all-one vector, the position embeddings $\boldsymbol{p}_n$ are set to zero, $\boldsymbol{W}_{K,Q,V} \in \mathbb{R}^{m \times d}$, and $\boldsymbol{W}_1 \in \mathbb{R}^{r \times d}$ can be set to any arbitrary value. This concludes the explicit construction of $\boldsymbol{\theta}_\star$ and the proof. $\qquad \square$

## B.2 PROOF OF THM. 1 FOR $p + q > 1$, WEIGHT-TYING

*Proof.* We use a similar idea as in the proof for the $p + q \le 1$ case by constructing a $\boldsymbol{\theta} \in \mathbb{R}^{D-d}$ satisfying $f_{\boldsymbol{\theta}}(x_1^n) = \mathbb{P}(x_{n+1} = 1 \mid x_n) = x_n(1 - p - q) + p$. However, in this case we need to use the ReLU component of the FF mechanism unlike the earlier case where we set $\boldsymbol{W}_2 = 0$. Now we start with constructing $\boldsymbol{\theta}_\star$.

Let the embedding $\boldsymbol{e} = \boldsymbol{a} = \mathbf{1}$ and the positional encoding $\boldsymbol{p}_n = -\frac{1}{2}\mathbf{1}$ for all $n \ge 1$, where $\mathbf{1} \in \mathbb{R}^d$ denotes the all-one vector. Thus $\boldsymbol{x}_n = \alpha_n \mathbf{1}$ with $\alpha_n = +\frac{1}{2}$ when $x_n = 1$ and $\alpha_n = -\frac{1}{2}$ when $x_n = 0$. Now let $\boldsymbol{W}_O = 0$ in the Attention layer. Hence $\boldsymbol{y}_n = \boldsymbol{x}_n = \alpha_n \mathbf{1}$. For the FF layer, let $\boldsymbol{W}_1$ and $\boldsymbol{W}_2$ be such that (to be determined later)

$$\boldsymbol{W}_2 \, \mathrm{ReLU}(\boldsymbol{W}_1 \, \boldsymbol{y}_n) = \beta_n \mathbf{1},$$

and hence

$$\boldsymbol{z}_n = \boldsymbol{x}_n + \boldsymbol{W}_2 \, \mathrm{ReLU}(\boldsymbol{W}_1 \, \boldsymbol{y}_n) = \alpha_n \mathbf{1} + \beta_n \mathbf{1} = (\alpha_n + \beta_n)\mathbf{1}.$$

Thus the logits are given by $\mathrm{logit}_n = \sigma(\langle \boldsymbol{a}, \boldsymbol{z}_n \rangle + b) = \sigma(d(\alpha_n + \beta_n) + b)$. Since $f_{\boldsymbol{\theta}}(x_1^n) = \sigma(\mathrm{logit}_n) = \mathbb{P}(x_{n+1} = 1 \mid x_n)$, we have that

$$\sigma(\mathrm{logit}_n) = \sigma(d(\alpha_n + \beta_n) + b) = x_n(1 - p - q) + p, \quad x_n \in \{0, 1\}.$$

Rewriting,

$$d(\alpha_n + \beta_n) + b = \log \left( \frac{x_n(1 - p - q) + p}{(1 - p)(1 - x_n) + q x_n} \right), \quad x_n \in \{0, 1\}.$$

Substituting $x_n = 0$ and $x_n = 1$, and denoting corresponding $\beta$'s by $\beta_1$ and $\beta_0$ (with a slight abuse of notation), we further simiplify to

$$d\left(-\frac{1}{2} + \beta_0\right) + b = \log\left(\frac{p}{1-p}\right), \tag{8}$$

$$d\left(\frac{1}{2} + \beta_1\right) + b = \log\left(\frac{1-q}{q}\right).$$

Subtracting both the above equations we obtain

$$d(1 + \beta_1 - \beta_0) = \underbrace{\log\left(\frac{(1-p)(1-q)}{pq}\right)}_{<0 \text{ when } p+q>1}. \tag{9}$$

Now it suffices to find $\beta_1$ and $\beta_0$ satisfying Eq. (9). Recall that $\beta_n$ obeys $\boldsymbol{W}_2 \operatorname{ReLU}(\boldsymbol{W}_1 \boldsymbol{y}_n) = \beta_n \mathbf{1}$. Let $\boldsymbol{W}_1 = w \, \mathbf{1}\mathbf{1}^\top$ and $\boldsymbol{W}_2 = -\boldsymbol{W}_1^\top$ for some $w \in \mathbb{R}$. Since $\boldsymbol{y}_n = \alpha_n \mathbf{1}$, we have

$$-w \, \mathbf{1}\mathbf{1}^\top \operatorname{ReLU}(w \, \mathbf{1}\mathbf{1}^\top \alpha_n \mathbf{1}) = \beta_n \mathbf{1}.$$

Simplifying,

$$\beta_n = -w^2 d \cdot \mathbf{1}^\top \operatorname{ReLU}(\alpha_n \mathbf{1}), \quad \alpha_n = \pm\frac{1}{2}.$$

Thus $\beta_0 = 0$ (corresponding to $x_n = 0$ and $\alpha_n = -\frac{1}{2}$) and $\beta_1 = -\frac{w^2 d^2}{2}$ (otherwise). Substituting them in Eq. (9), we have

$$1 - \frac{w^2 d^2}{2} = \frac{1}{d} \cdot \log\left(\frac{(1-p)(1-q)}{pq}\right).$$

Let $w = w_\star$ be a solution to the above equation, i.e.

$$w_\star = \sqrt{\frac{2}{d^2}\left(1 - \frac{1}{d} \cdot \log\left(\frac{(1-p)(1-q)}{pq}\right)\right)}.$$

By substituting $\beta_0 = 0$ in Eq. (8) we obtain the bias $b_\star = \log\left(\frac{p}{1-p}\right) + \frac{d}{2}$. Piecing everything together, let

$$\boldsymbol{\theta}_\star = \left(\boldsymbol{e} = \boldsymbol{a} = \mathbf{1}, \{\boldsymbol{p}_n = (-1/2)\,\mathbf{1}\}_{n=1}^N, \boldsymbol{W}_O = 0, \boldsymbol{W}_{K,Q,V}, \boldsymbol{W}_1 = w_\star \, \mathbf{1}\mathbf{1}^\top, \boldsymbol{W}_2 = -\boldsymbol{W}_1^\top, b = b_\star\right), \tag{10}$$

and we are done.

$\square$

### B.3 PROOF OF THM. 1 FOR NON-WEIGHT-TIED

*Proof.* By extending $\boldsymbol{\theta}_\star \in \mathbb{R}^{D-d}$ to $\bar{\boldsymbol{\theta}}_\star \triangleq (\boldsymbol{\theta}_\star, \boldsymbol{a}_\star) \in \mathbb{R}^D$, it follows from the Transformer architecture in § A that $\mathbb{P}_{\bar{\boldsymbol{\theta}}_\star}(x_{n+1} = 1 \mid x_1^n) = \mathbb{P}_{\boldsymbol{\theta}_\star}(x_{n+1} = 1 \mid x_1^n) = \mathbb{P}(x_{n+1} = 1 \mid x_n)$, the Markov kernel. As the proof of Thm. 1 in § B.2 and § B.1 establish, prediction probability equalling the kernel is a sufficient condition for global-optimality. Hence $\bar{\boldsymbol{\theta}}_\star$ is a global minimum for $L(\cdot)$ in $\mathbb{R}^D$. $\square$

### B.4 PROOF OF THM. 2

*Proof.* First we construct an explicit $\boldsymbol{\theta}_{\boldsymbol{\pi}} \in \mathbb{R}^{D-d}$ such that it satifies properties (ii)–(iv) of Thm. 2 i.e. it is a stationary point with loss value being the entropy of the marginal $H(\boldsymbol{\pi})$ and that it captures the marginal distribution $\mathbb{P}(x_{n+1} = 1) = \pi_1$. Then we compute its Hessian and show that it is a local minimum for $p + q > 1$ thus proving property (i). On the other hand, the same $\boldsymbol{\theta}_{\boldsymbol{\pi}}$ could either be a local minimum or saddle point for $p + q < 1$. We start with the construction.

Recall that the full set of the Transformer parameters in the weight-tied case is given by $\boldsymbol{\theta} = (\boldsymbol{e} = \boldsymbol{a}, \{\boldsymbol{p}_n\}_{n=1}^N, \boldsymbol{W}_O, \boldsymbol{W}_{K,Q,V}, \boldsymbol{W}_2, \boldsymbol{W}_1, b) \in \mathbb{R}^{D-d}$. Define $\boldsymbol{\theta}_{\boldsymbol{\pi}} \in \mathbb{R}^{D-d}$ to be

$$\boldsymbol{\theta}_{\boldsymbol{\pi}} = \left( \boldsymbol{e} = \boldsymbol{a} = 0, \{\boldsymbol{p}_n\}_{n=1}^N, \boldsymbol{W}_O = 0, \boldsymbol{W}_{K,Q,V}, \boldsymbol{W}_2 = 0, \boldsymbol{W}_1, b = \log\left(\frac{p}{q}\right) \right), \quad (11)$$

where $\{\boldsymbol{p}_n\}_{n=1}^N \subset \mathbb{R}^d$, $\boldsymbol{W}_{K,Q,V} \in \mathbb{R}^{m \times d}$, and $\boldsymbol{W}_1 \in \mathbb{R}^{r \times d}$ can be set to any arbitrary value. Now we start with property (ii).

(ii): $f_{\boldsymbol{\theta}_{\boldsymbol{\pi}}}(x_1^n) = \mathbb{P}_{\boldsymbol{\theta}_{\boldsymbol{\pi}}}(x_{n+1} = 1 \mid x_1^n) = \mathbb{P}(x_{n+1} = 1) = \pi_1$.

Since $\boldsymbol{a} = 0$, it follows from (Linear) and (Prediction) layers that $f_{\boldsymbol{\theta}_{\boldsymbol{\pi}}}(x_1^n) = \sigma(b) = \sigma(\log(p/q)) = \frac{p}{p+q} = \pi_1$. In other words, the model ignores all the inputs and outputs a constant probability $\pi_1$.

(iii): $L(\boldsymbol{\theta}_{\boldsymbol{\pi}}) = H(x_{n+1}) = H(\boldsymbol{\pi})$.

Since $f_{\boldsymbol{\theta}_{\boldsymbol{\pi}}}(\cdot) = \pi_1 = \mathbb{E}[x_{n+1}]$, it follows from Eq. (3) that

$$L(\boldsymbol{\theta}_{\boldsymbol{\pi}}) = -\frac{1}{N} \sum_{n \in [N]} \mathbb{E}_{x_1^{n+1}}[x_{n+1} \cdot \log f_{\boldsymbol{\theta}_{\boldsymbol{\pi}}}(x_1^n) + (1 - x_{n+1}) \cdot \log(1 - f_{\boldsymbol{\theta}_{\boldsymbol{\pi}}}(x_1^n))]$$

$$= -\frac{1}{N} \sum_{n \in [N]} \mathbb{E}_{x_1^{n+1}}[x_{n+1} \cdot \log \pi_1 + (1 - x_{n+1}) \cdot \log \pi_0]$$

$$= \frac{1}{N} \sum_{n \in [N]} [-\pi_1 \log \pi_1 - \pi_0 \log \pi_0]$$

$$= H(\boldsymbol{\pi}) = H(x_{n+1}).$$

(iv): $\nabla L(\boldsymbol{\theta}_{\boldsymbol{\pi}}) = 0$.

At $\boldsymbol{\theta} = \boldsymbol{\theta}_{\boldsymbol{\pi}}$, the individual layer outputs of the Transformer (§ A) are given by

$$x_n \in \{0, 1\} \xrightarrow{\text{Uni-embedding}} \boldsymbol{x}_n = \boldsymbol{p}_n \xrightarrow{\text{Attention}} \boldsymbol{y}_n = \boldsymbol{p}_n \xrightarrow{\text{FF}} \boldsymbol{z}_n = \boldsymbol{p}_n \xrightarrow{\text{Linear}} \text{logit}_n = b \xrightarrow{\text{Prediction}} f_{\boldsymbol{\theta}_{\boldsymbol{\pi}}}(x_1^n) = \pi_1.$$

In other words, none of the layer outputs depend on the input sequence $\{x_n\}_{n=1}^N$. In view of this fact and $\mathbb{E}[x_{n+1}] = \pi_1$, using Lemma 2 the gradient with respect to $\boldsymbol{a}$ of $L$ at $\boldsymbol{\theta} = \boldsymbol{\theta}_{\boldsymbol{\pi}}$ is given by

$$\nabla_{\boldsymbol{a}} L = -\frac{1}{N} \sum_{n \in [N]} \mathbb{E}_{x_1^{n+1}}\left[(x_{n+1} - f_{\boldsymbol{\theta}_{\boldsymbol{\pi}}}(x_1^n)) \cdot \nabla_{\boldsymbol{a}}\left(\boldsymbol{a}^\top \boldsymbol{z}_n + b\right)\right]$$

$$= -\frac{1}{N} \sum_{n \in [N]} \mathbb{E}_{x_1^{n+1}}\left[(x_{n+1} - \pi_1)(\boldsymbol{z}_n + \nabla_{\boldsymbol{a}} \boldsymbol{z}_n \cdot \boldsymbol{a})\right]$$

$$\overset{(\boldsymbol{a}=0)}{=} -\frac{1}{N} \sum_{n \in [N]} \mathbb{E}_{x_{n+1}}\left[(x_{n+1} - \pi_1) \cdot \boldsymbol{p}_n\right]$$

$$= 0.$$

Similarly, for $b$:

$$\nabla_b L = -\frac{1}{N} \sum_{n \in [N]} \mathbb{E}_{x_1^{n+1}}\left[(x_{n+1} - f_{\boldsymbol{\theta}_{\boldsymbol{\pi}}}(x_1^n)) \cdot \nabla_b\left(\boldsymbol{a}^\top \boldsymbol{z}_n + b\right)\right] = -\frac{1}{N} \sum_{n \in [N]} \mathbb{E}_{x_{n+1}}\left[x_{n+1} - \pi_1\right] = 0.$$

For any other parameter $\boldsymbol{w} \in \boldsymbol{\theta}$ apart from $\boldsymbol{a}, b$, we see from Eq. (5) that the gradient $\nabla_{\boldsymbol{w}} L$ has the term $\nabla_{\boldsymbol{w}}(\boldsymbol{a}^\top \boldsymbol{z}_n) = (\nabla_{\boldsymbol{w}} \boldsymbol{z}_n) \cdot \boldsymbol{a}$ inside the expectation $\mathbb{E}_{x_1^{n+1}}[\ldots]$. Since $\boldsymbol{a} = 0$, this equals zero and hence $\nabla_{\boldsymbol{w}} L = 0$.

Together $\nabla L(\boldsymbol{\theta}_{\boldsymbol{\pi}}) = 0$.

(i): $\boldsymbol{\theta}_{\boldsymbol{\pi}}$ is a bad local minimum for $L$ when $p + q > 1$.

Towards establishing this, we first let $\boldsymbol{\alpha} = (b, \boldsymbol{a})$ and $\boldsymbol{\beta} = (\{\boldsymbol{p}_n\}_{n=1}^N, \boldsymbol{W}_O, \boldsymbol{W}_{K,Q,V}, \boldsymbol{W}_2, \boldsymbol{W}_1)$ be two different sets of parameters comprising $\boldsymbol{\theta}$, i.e. $\boldsymbol{\theta} = (\boldsymbol{\alpha}, \boldsymbol{\beta})$ and compute the Hessian $\boldsymbol{H}_{\boldsymbol{\pi}} \triangleq$

$\nabla^{(2)} L(\boldsymbol{\theta})|_{\boldsymbol{\theta}=\boldsymbol{\theta}_{\boldsymbol{\pi}}}$ and show that it has the following block-diagonal structure:

$$\boldsymbol{H}_{\boldsymbol{\pi}} = \begin{bmatrix} \boldsymbol{H}_{\boldsymbol{\alpha}} & 0 \\ 0 & 0 \end{bmatrix},$$

where $\boldsymbol{H}_{\boldsymbol{\alpha}}$ corresponds to the Hessian with respect to the parameters $\boldsymbol{a}$ and $b$ in $\boldsymbol{\alpha}$. Further we show that if $p + q > 1$, $\boldsymbol{H}_{\boldsymbol{\alpha}} \succ 0$ i.e. it is positive-definite. This helps us in establishing that $\boldsymbol{\theta}_{\boldsymbol{\pi}}$ a local minimum. Now we start with the Hessian computation.

**Hessian computation.** We first compute the Hessian with respect to $\boldsymbol{\alpha}$.

From Lemma 2, we have that second derivative with respect to $b$ at $\boldsymbol{\theta} = \boldsymbol{\theta}_{\boldsymbol{\pi}}$ is given by

$$
\begin{aligned}
\nabla_b^{(2)} L = \nabla_b \left( \nabla_b L \right) &= \nabla_b \left( -\frac{1}{N} \sum_{n \in [N]} \mathbb{E}_{x_1^{n+1}} \left[ x_{n+1} - f_{\boldsymbol{\theta}}(x_1^n) \right] \right) \\
&\overset{(a)}{=} \frac{1}{N} \sum_{n \in [N]} \mathbb{E} \left[ f_{\boldsymbol{\theta}}(x_1^n)(1 - f_{\boldsymbol{\theta}}(x_1^n)) \cdot \nabla_b \left( \boldsymbol{a}^\top \boldsymbol{z}_n + b \right) \right] \\
&\overset{(\boldsymbol{\theta}=\boldsymbol{\theta}_{\boldsymbol{\pi}})}{=} \frac{1}{N} \sum_{n \in [N]} \mathbb{E}[\pi_1 \pi_0] \\
&= \pi_0 \pi_1 > 0,
\end{aligned}
$$

where $(a)$ follows from the fact that $\nabla_b f_{\boldsymbol{\theta}}(x_1^n) = \nabla_b \sigma(\boldsymbol{a}^\top \boldsymbol{z}_n + b) = f_{\boldsymbol{\theta}}(x_1^n)(1 - f_{\boldsymbol{\theta}}(x_1^n)) \cdot \nabla_b \left( \boldsymbol{a}^\top \boldsymbol{z}_n + b \right)$. Now we compute the second derivative with respect to $\boldsymbol{a}$. From Lemma 2, we obtain

$$
\begin{aligned}
\nabla_{\boldsymbol{a}}^{(2)} L = \nabla_{\boldsymbol{a}} \left( \nabla_{\boldsymbol{a}} L \right) &= \nabla_{\boldsymbol{a}} \left( -\frac{1}{N} \sum_{n \in [N]} \mathbb{E}_{x_1^{n+1}} \left[ (x_{n+1} - f_{\boldsymbol{\theta}}(x_1^n)) \cdot \nabla_{\boldsymbol{a}}(\boldsymbol{a}^\top \boldsymbol{z}_n) \right] \right) \\
&= \nabla_{\boldsymbol{a}} \left( -\frac{1}{N} \sum_{n \in [N]} \mathbb{E} \left[ (x_{n+1} - f_{\boldsymbol{\theta}}(x_1^n))(\boldsymbol{z}_n + (\nabla_{\boldsymbol{a}} \boldsymbol{z}_n) \cdot \boldsymbol{a}) \right] \right) \\
&\overset{(a)}{=} \frac{1}{N} \sum_{n \in [N]} \mathbb{E} \left[ f_{\boldsymbol{\theta}}(1 - f_{\boldsymbol{\theta}})(\boldsymbol{z}_n + (\nabla_{\boldsymbol{a}} \boldsymbol{z}_n) \cdot \boldsymbol{a})(\boldsymbol{z}_n + (\nabla_{\boldsymbol{a}} \boldsymbol{z}_n) \cdot \boldsymbol{a})^\top \right] \\
&\quad - \frac{1}{N} \sum_{n \in [N]} \mathbb{E} \left[ (x_{n+1} - f_{\boldsymbol{\theta}}(x_1^n))(2 \nabla_{\boldsymbol{a}} \boldsymbol{z}_n) \right],
\end{aligned}
$$

where $(a)$ follows from the gradient of the product rule and the fact that $\nabla_{\boldsymbol{a}} f_{\boldsymbol{\theta}}(x_1^n) = f_{\boldsymbol{\theta}}(x_1^n)(1 - f_{\boldsymbol{\theta}}(x_1^n))(\boldsymbol{z}_n + (\nabla_{\boldsymbol{a}} \boldsymbol{z}_n) \cdot \boldsymbol{a})$. At $\boldsymbol{\theta} = \boldsymbol{\theta}_{\boldsymbol{\pi}}$, this further simplifies to

$$
\begin{aligned}
\nabla_{\boldsymbol{a}}^{(2)} L &\overset{(b)}{=} \frac{1}{N} \sum_{n \in [N]} \left( \mathbb{E}[\pi_1 \pi_0 \cdot \boldsymbol{p}_n \boldsymbol{p}_n^\top] - 2\mathbb{E}[(x_{n+1} - \pi_1)x_n \boldsymbol{I}] \right) \\
&\overset{(c)}{=} \frac{1}{N} \sum_{n \in [N]} \left( (\pi_0 \pi_1) \cdot \boldsymbol{p}_n \boldsymbol{p}_n^\top \right) - 2\mathbb{E}[(x_n(1 - p - q) + p - \pi_1)x_n \boldsymbol{I}] \\
&= \frac{1}{N} \sum_{n \in [N]} \left( (\pi_0 \pi_1) \cdot \boldsymbol{p}_n \boldsymbol{p}_n^\top \right) - 2\mathbb{E}[x_n(\pi_0 - q)\boldsymbol{I}] \\
&= \frac{1}{N} \sum_{n \in [N]} \left( (\pi_0 \pi_1) \cdot \boldsymbol{p}_n \boldsymbol{p}_n^\top \right) - 2\pi_1(\pi_0 - q)\boldsymbol{I}] \\
&= \pi_0 \pi_1 \left( \sum_{n \in [N]} \frac{\boldsymbol{p}_n \boldsymbol{p}_n^\top}{N} - 2\left(1 - \frac{q}{\pi_0}\right) \boldsymbol{I} \right)
\end{aligned}
$$

$$\stackrel{(d)}{=} \pi_0 \pi_1 \left( \sum_{n \in [N]} \frac{\boldsymbol{p}_n \boldsymbol{p}_n^\top}{N} + 2(p + q - 1) \boldsymbol{I} \right),$$

where $(b)$ follows from the fact that $\nabla_{\boldsymbol{a}} \boldsymbol{z}_n = x_n \boldsymbol{I}$ at $\boldsymbol{\theta} = \boldsymbol{\theta}_{\boldsymbol{\pi}}$ where $\boldsymbol{I}$ is the identity matrix is $\mathbb{R}^{d \times d}$, $(c)$ from the observation that $\mathbb{E}[x_{n+1}|x_n] = x_n(1 - p - 1) + p$, and $(d)$ from the fact that $\pi_0 = \frac{q}{p+q}$. Now we compute the cross-derivative of second order $\nabla_{\boldsymbol{a}, b} L$. Again, invoking Lemma 2,

$$\nabla_{\boldsymbol{a}b} L = \nabla_{\boldsymbol{a}}(\nabla_b L) = \nabla_{\boldsymbol{a}} \left( -\frac{1}{N} \sum_{n \in [N]} \mathbb{E}_{x_1^{n+1}} [x_{n+1} - f_{\boldsymbol{\theta}}(x_1^n)] \right)$$

$$= \frac{1}{N} \sum_{n \in [N]} \mathbb{E} \left[ f_{\boldsymbol{\theta}}(1 - f_{\boldsymbol{\theta}})(\boldsymbol{z}_n + (\nabla_{\boldsymbol{a}} \boldsymbol{z}_n) \cdot \boldsymbol{a}) \right]$$

$$\stackrel{(\boldsymbol{\theta}=\boldsymbol{\theta}_{\boldsymbol{\pi}})}{=} \frac{1}{N} \sum_{n \in [N]} \mathbb{E} \left[ \pi_1 \pi_0 \cdot \boldsymbol{p}_n \right]$$

$$= \pi_0 \pi_1 \left( \sum_{n \in [N]} \frac{\boldsymbol{p}_n}{N} \right).$$

Piecing all the results together we obtain that for $\boldsymbol{\alpha} = (b, \boldsymbol{a})$, its corresponding Hessian is given by

$$\boldsymbol{H}_{\boldsymbol{\alpha}} \triangleq \nabla_{\boldsymbol{\alpha}}^{(2)} L(\boldsymbol{\alpha}_{\boldsymbol{\pi}}, \boldsymbol{\beta}_{\boldsymbol{\pi}}) = \pi_0 \pi_1 \begin{bmatrix} 1 & \boldsymbol{u}^\top \\ \boldsymbol{u} & \boldsymbol{V} \end{bmatrix}, \quad \boldsymbol{u} \triangleq \sum_{n \in [N]} \frac{\boldsymbol{p}_n}{N}, \boldsymbol{V} \triangleq \sum_{n \in [N]} \frac{\boldsymbol{p}_n \boldsymbol{p}_n^\top}{N} + 2(p + q - 1) \boldsymbol{I}.$$

$$(12)$$

We now show that the Hessian $\boldsymbol{H}_{\boldsymbol{\beta}} \triangleq \nabla_{\boldsymbol{\beta}}^{(2)} L(\boldsymbol{\alpha}_{\boldsymbol{\pi}}, \boldsymbol{\beta}_{\boldsymbol{\pi}}) = 0$. Recall that $\boldsymbol{\beta} = (\{\boldsymbol{p}_n\}_{n=1}^N, \boldsymbol{W}_O, \boldsymbol{W}_{K,Q,V}, \boldsymbol{W}_2, \boldsymbol{W}_1)$. For any $\boldsymbol{w}_1, \boldsymbol{w}_2 \in \boldsymbol{\beta}$, Lemma 2 implies that

$$\nabla_{\boldsymbol{w}_1 \boldsymbol{w}_2} L = \nabla_{\boldsymbol{w}_1} (\nabla_{\boldsymbol{w}_2} L) = \nabla_{\boldsymbol{w}_1} \left( -\frac{1}{N} \sum_{n \in [N]} \mathbb{E} \left[ (x_{n+1} - f_{\boldsymbol{\theta}}(x_1^n)(\nabla_{\boldsymbol{w}_2} \boldsymbol{z}_n \cdot \boldsymbol{a})) \right] \right)$$

$$\stackrel{(a)}{=} \frac{1}{N} \sum_{n \in [N]} \mathbb{E} \left[ f_{\boldsymbol{\theta}}(1 - f_{\boldsymbol{\theta}})(\nabla_{\boldsymbol{w}_2} \boldsymbol{z}_n \cdot \boldsymbol{a})(\nabla_{\boldsymbol{w}_1} \boldsymbol{z}_n \cdot \boldsymbol{a})^\top \right]$$

$$\stackrel{(\boldsymbol{\theta}=\boldsymbol{\theta}_{\boldsymbol{\pi}})}{=} 0,$$

where $(a)$ follows from the fact that $\nabla_{\boldsymbol{w}_1} f_{\boldsymbol{\theta}}(x_1^n) = \nabla_{\boldsymbol{w}_1} \sigma(\boldsymbol{a}^\top \boldsymbol{z}_n + b) = f_{\boldsymbol{\theta}}(x_1^n)(1 - f_{\boldsymbol{\theta}}(x_1^n))(\nabla_{\boldsymbol{w}_1} \boldsymbol{z}_n \cdot \boldsymbol{a})$. Thus $\boldsymbol{H}_{\boldsymbol{\beta}} = 0$. Similarly, we can show that $\boldsymbol{H}_{\boldsymbol{\alpha}\boldsymbol{\beta}} = \nabla_{\boldsymbol{\alpha}\boldsymbol{\beta}} L = 0$ and hence $\boldsymbol{H}_{\boldsymbol{\beta}\boldsymbol{\alpha}} = \boldsymbol{H}_{\boldsymbol{\alpha}\boldsymbol{\beta}}^\top = 0$. Thus,

$$\boldsymbol{H}_{\boldsymbol{\pi}} = \nabla^{(2)} L(\boldsymbol{\theta}_{\boldsymbol{\pi}}) = \begin{bmatrix} \boldsymbol{H}_{\boldsymbol{\alpha}} & 0 \\ 0 & 0 \end{bmatrix}$$

Now it remains to show that $\boldsymbol{H}_{\boldsymbol{\alpha}}$ is positive-definite when $p + q > 1$ and it implies that $\boldsymbol{\theta}_{\boldsymbol{\pi}}$ is a local minimum.

**Positive-definitenss of $\boldsymbol{H}_{\boldsymbol{\alpha}}$.** Recall from Eq. (12) that $\boldsymbol{H}_{\boldsymbol{\alpha}} = \begin{bmatrix} 1 & \boldsymbol{u}^\top \\ \boldsymbol{u} & \boldsymbol{V} \end{bmatrix}$, where $\boldsymbol{u} = \sum_{n \in [N]} \boldsymbol{p}_n / N, \boldsymbol{V} = \sum_{n \in [N]} \boldsymbol{p}_n \boldsymbol{p}_n^\top / N + 2(p + q - 1) \boldsymbol{I}$. From the characterization of positive-definiteness by Schur's complement Horn and Johnson (2012), we have that $\boldsymbol{H}_{\boldsymbol{\alpha}} \succ 0 \Leftrightarrow 1 > 0$ and $\boldsymbol{V} - \boldsymbol{u} \boldsymbol{u}^\top \succ 0$. We have that

$$\boldsymbol{V} - \boldsymbol{u} \boldsymbol{u}^\top = 2(p + q - 1) \boldsymbol{I} + \sum_{n \in [N]} \frac{\boldsymbol{p}_n \boldsymbol{p}_n^\top}{N} - \left( \sum_{n \in [N]} \frac{\boldsymbol{p}_n}{N} \right) \left( \sum_{n \in [N]} \frac{\boldsymbol{p}_n}{N} \right)^\top$$

$$= 2(p+q-1)\boldsymbol{I} + \sum_{n\in[N]} \frac{1}{N} \left( \boldsymbol{p}_n - \sum_{n\in[N]} \frac{\boldsymbol{p}_n}{N} \right) \left( \boldsymbol{p}_n - \sum_{n\in[N]} \frac{\boldsymbol{p}_n}{N} \right)^\top$$

$$= 2(p+q-1)\boldsymbol{I} + \mathrm{Cov}(\{\boldsymbol{p}_n\}_{n=1}^N), \tag{13}$$

where $\mathrm{Cov}(\{\boldsymbol{p}_n\}_{n=1}^N) = \sum_{n\in[N]} \frac{1}{N} \left( \boldsymbol{p}_n - \sum_{n\in[N]} \frac{\boldsymbol{p}_n}{N} \right) \left( \boldsymbol{p}_n - \sum_{n\in[N]} \frac{\boldsymbol{p}_n}{N} \right)^\top$ is the covariance matrix of the set $\{\boldsymbol{p}_n\}_{n=1}^N$ and hence positive semi-definite. Thus if $p+q>1$, we have that $2(p+q-1)\boldsymbol{I} \succ 0$ and together, we obtain that $\boldsymbol{V} - \boldsymbol{u}\boldsymbol{u}^\top \succ 0$. Hence $\boldsymbol{H}_{\boldsymbol{\alpha}} \succ 0$. Now it remains to show that $\boldsymbol{\theta}_{\boldsymbol{\pi}}$ is a local minimum.

**$\boldsymbol{H}_{\boldsymbol{\alpha}}$ is positive-definite implies $\boldsymbol{\theta}_{\boldsymbol{\pi}}$ is a local minimum.** Since $\boldsymbol{H}_{\boldsymbol{\alpha}} \succ 0$, let $\boldsymbol{H}_{\boldsymbol{\alpha}} \succcurlyeq \lambda \boldsymbol{I}$ for some $\lambda > 0$ (in fact $\lambda = 2(p+q-1)$ works). Since $\boldsymbol{\theta} = (\boldsymbol{\alpha}, \boldsymbol{\beta}) \in \mathbb{R}^{D-d}$, interpret $L(\boldsymbol{\theta}) = L(\boldsymbol{\alpha}, \boldsymbol{\beta})$ as a function of two variables $\boldsymbol{\alpha}$ and $\boldsymbol{\beta}$ with appropriate dimensions. We know the following facts about $L(\cdot, \cdot)$:

- **Fact 1.** $\boldsymbol{\alpha} \mapsto L(\boldsymbol{\alpha}, \boldsymbol{\beta}_{\boldsymbol{\pi}})$ has a local minimum (as a function of one variable) at $\boldsymbol{\alpha} = \boldsymbol{\alpha}_{\boldsymbol{\pi}}$ (since $\boldsymbol{H}_{\boldsymbol{\alpha}} \succ 0$).

- **Fact 2.** $\boldsymbol{\beta} \mapsto L(\boldsymbol{\alpha}_{\boldsymbol{\pi}}, \boldsymbol{\beta})$ is constant in $\boldsymbol{\beta}$ (since $\boldsymbol{a}_{\boldsymbol{\pi}} = 0$, the probability $f_{\boldsymbol{\theta}}(x_1^n)$ is constant w.r.t. $\boldsymbol{z}_n$ and hence w.r.t. $\boldsymbol{\beta}$ (Linear)).

- **Fact 3.** $\nabla L(\boldsymbol{\alpha}_{\boldsymbol{\pi}}, \boldsymbol{\beta}_{\boldsymbol{\pi}}) = 0$ and $\boldsymbol{H}_{\boldsymbol{\pi}} = \nabla^{(2)} L(\boldsymbol{\alpha}_{\boldsymbol{\pi}}, \boldsymbol{\beta}_{\boldsymbol{\pi}}) = \begin{bmatrix} \boldsymbol{H}_{\boldsymbol{\alpha}} & 0 \\ 0 & 0 \end{bmatrix}$ with $\boldsymbol{H}_{\boldsymbol{\alpha}} \succcurlyeq \lambda \boldsymbol{I}$.

Using these facts now we show that $(\boldsymbol{\alpha}_{\boldsymbol{\pi}}, \boldsymbol{\beta}_{\boldsymbol{\pi}}) = \boldsymbol{\theta}_{\boldsymbol{\pi}}$ is also a local minimum in two-variables. We prove this by contradiction. Suppose that $(\boldsymbol{\alpha}_{\boldsymbol{\pi}}, \boldsymbol{\beta}_{\boldsymbol{\pi}})$ is not a local minimum for $L(\cdot, \cdot)$. Without loss of generality, by a shift of cordinates treat $\boldsymbol{\theta}_{\boldsymbol{\pi}}$ as the origin, i.e. $(\boldsymbol{\alpha}_{\boldsymbol{\pi}} = 0, \boldsymbol{\beta}_{\boldsymbol{\pi}} = 0)$ is not a local minimum for $L(\cdot, \cdot)$. Then there exists an unit direction $\boldsymbol{d} = (\boldsymbol{u}, \boldsymbol{v}) \in \mathbb{R}^{D-d}$ with $\|\boldsymbol{d}\|^2 = \|\boldsymbol{u}\|^2 + \|\boldsymbol{v}\|^2 = 1$ and an $0 < \varepsilon_0 < 1$ such that

$$L(\varepsilon \boldsymbol{d}) < L(0), \quad \forall\, 0 < \varepsilon \le \varepsilon_0 < 1. \tag{14}$$

Clearly $\|\boldsymbol{u}\| > 0$, otherwise it will contradict Fact 2. Using the definition of directional-derivative, we have that

$$\boldsymbol{d}^\top \nabla^{(2)} L(0, 0) \boldsymbol{d} = \lim_{\varepsilon \to 0} \frac{\langle \nabla L(\varepsilon \boldsymbol{d}), \boldsymbol{d} \rangle - \langle \nabla L(0), \boldsymbol{d} \rangle}{\varepsilon}$$

$$= \lim_{\varepsilon \to 0} \frac{\langle \nabla L(\varepsilon \boldsymbol{d}), \boldsymbol{d} \rangle}{\varepsilon}.$$

On the other hand, using the Hessian structure the LHS equals $\boldsymbol{d}^\top \nabla^{(2)} L(0, 0) \boldsymbol{d} \ge \lambda \|\boldsymbol{u}\|^2 \triangleq K_1 > 0$. Thus

$$\lim_{\varepsilon \to 0} \frac{\langle \nabla L(\varepsilon \boldsymbol{d}), \boldsymbol{d} \rangle}{\varepsilon} = K_1 > 0.$$

Thus there exists an $\varepsilon_1 > 0$ and $K > 0$ such that

$$\frac{\langle \nabla L(\varepsilon \boldsymbol{d}), \boldsymbol{d} \rangle}{\varepsilon} \ge K, \quad \forall\, 0 < \varepsilon \le \varepsilon_1,$$

which implies

$$\langle \nabla L(\varepsilon \boldsymbol{d}), \boldsymbol{d} \rangle \ge K\varepsilon, \quad \forall\, 0 \le \varepsilon \le \varepsilon_1.$$

Defining the function $g : \mathbb{R}_+ \to \mathbb{R}$ as $g(\varepsilon) = L(\varepsilon \boldsymbol{d})$, we obtain that $g'(\varepsilon) = \langle \nabla L(\varepsilon \boldsymbol{d}), \boldsymbol{d} \rangle \ge K\varepsilon$ for $0 \le \varepsilon \le \varepsilon_1$. Using the fundamental theorem of Calculus, we have that for any $0 \le \varepsilon \le \varepsilon_1$,

$$g(\varepsilon) - g(0) = \int_0^\varepsilon g'(t)\,dt$$

$$\ge \int_0^\varepsilon K t\,dt$$

$$= \frac{K\varepsilon^2}{2}.$$

Thus $g(\varepsilon) = L(\varepsilon \boldsymbol{d}) \geq L(0) + \frac{K\varepsilon^2}{2}$ for all $0 \leq \varepsilon \leq \varepsilon_1$ whereas $L(\varepsilon \boldsymbol{d}) < L(0)$ for all $0 < \varepsilon < \varepsilon_0$ from Eq. (14). Choosing $\varepsilon_\star = \min(\varepsilon_0, \varepsilon_1)$, we have a contradiction for $0 < \varepsilon < \varepsilon_\star$. Thus $0 \equiv \boldsymbol{\theta_\pi} = (\boldsymbol{\alpha_\pi}, \boldsymbol{\beta_\pi})$ is a local minimum. $\qquad\square$

## B.5 PROOF OF THM. 3

*Proof.* Since $\bar{\boldsymbol{\theta}}_{\boldsymbol{\pi}} \triangleq (\boldsymbol{\theta_\pi}, \boldsymbol{a_\pi}) \in \mathbb{R}^D$ is a canonical extension of $\boldsymbol{\theta_\pi} = (\boldsymbol{e_\pi} = \boldsymbol{a_\pi}, \ldots, \boldsymbol{b_\pi}) \in \mathbb{R}^{D-d}$, which is a local-minimum for $L(\cdot)$ in $\mathbb{R}^{D-d}$, following the same-steps for the gradient computation and probability evaluation as in proof of Thm. 2 in § B.4, it immediately follows that $\bar{\boldsymbol{\theta}}_{\boldsymbol{\pi}}$ also satisfies properties (ii)-(iv), i.e. it's a stationary point, it captures the marginal, since $\mathbb{P}_{\bar{\boldsymbol{\theta}}_{\boldsymbol{\pi}}}(x_{n+1} = 1 \mid x_1^n) = \mathbb{P}_{\boldsymbol{\theta_\pi}}(x_{n+1} = 1 \mid x_1^n) = \mathbb{P}(x_{n+1} = 1) = \pi$, and hence its loss equals entropy of stationary distribution $H(\boldsymbol{\pi})$. In a similar fashion, the Hessian computation is essentially the same except for a slight difference in the Hessian structure, i.e.

$$\boldsymbol{H}(\bar{\boldsymbol{\theta}}_{\boldsymbol{\pi}}) \triangleq \nabla^{(2)} L(\bar{\boldsymbol{\theta}}_{\boldsymbol{\pi}}) = \begin{bmatrix} \boldsymbol{H_\alpha} & 0 \\ 0 & 0 \end{bmatrix}, \quad \boldsymbol{H_\alpha} = \pi_0 \pi_1 \begin{bmatrix} 1 & \boldsymbol{u}^\top & 0 \\ \boldsymbol{u} & \boldsymbol{V} & (p+q-1)\boldsymbol{I} \\ 0 & (p+q-1)\boldsymbol{I} & 0 \end{bmatrix},$$

where $\boldsymbol{u} \triangleq \sum_{n\in[N]} \frac{\boldsymbol{p}_n}{N}, \boldsymbol{V} \triangleq \sum_{n\in[N]} \frac{\boldsymbol{p}_n \boldsymbol{p}_n^\top}{N}$. In the weight-tied case, we observe that the matrix $\boldsymbol{V}$ also contains the $(p+q-1)\boldsymbol{I}$ terms which in the non-weight-tied case gets de-coupled (due to separate $\boldsymbol{e}$ and $\boldsymbol{a}$ parameters). In fact, $\boldsymbol{H_\alpha}$ corresponds to the Hessian w.r.t the parameters $\boldsymbol{\alpha} = (b, \boldsymbol{a}, \boldsymbol{e})$, i.e. $\boldsymbol{H_\alpha} = \nabla_\alpha^{(2)} L|_{\boldsymbol{\alpha}=\boldsymbol{\alpha_\pi}}$. Now it remains to show that $\boldsymbol{H_\alpha}$ is indefinite and hence $\bar{\boldsymbol{\theta}}_{\boldsymbol{\pi}}$ a saddle point.

Clearly, $\boldsymbol{H_\alpha}$ cannot be negative definite since with $\boldsymbol{d} = (1, 0, \ldots, 0)$, we have $\boldsymbol{d}^\top \boldsymbol{H_\alpha} \boldsymbol{d} = \pi_0 \pi_1 > 0$ for $(p,q) \in (0,1)$. Now we show that it cannot be positive definite either. Denoting

$$\boldsymbol{H_\alpha} = \pi_0 \pi_1 \begin{bmatrix} 1 & \boldsymbol{b}^\top \\ \boldsymbol{b} & \boldsymbol{C} \end{bmatrix}, \quad \boldsymbol{b} \triangleq \begin{bmatrix} \sum_{n\in[N]} \frac{\boldsymbol{p}_n}{N} \\ 0 \end{bmatrix}, \quad \boldsymbol{C} \triangleq \begin{bmatrix} \sum_{n\in[N]} \frac{\boldsymbol{p}_n \boldsymbol{p}_n^\top}{N} & (p+q-1)\boldsymbol{I} \\ (p+q-1)\boldsymbol{I} & 0 \end{bmatrix}.$$

Using the characterization of positive-definiteness by Schur's complement Horn and Johnson (2012), we have that $\boldsymbol{H_\alpha} \succ 0 \Leftrightarrow 1 > 0$ and $\boldsymbol{C} - \boldsymbol{b}\boldsymbol{b}^\top \succ 0$. This can be further simplified to

$$\boldsymbol{M} \triangleq \boldsymbol{C} - \boldsymbol{b}\boldsymbol{b}^\top = \begin{bmatrix} \sum_{n\in[N]} \frac{\boldsymbol{p}_n \boldsymbol{p}_n^\top}{N} & (p+q-1)\boldsymbol{I} \\ (p+q-1)\boldsymbol{I} & 0 \end{bmatrix} - \begin{bmatrix} \sum_{n\in[N]} \frac{\boldsymbol{p}_n}{N} \\ 0 \end{bmatrix} \begin{bmatrix} \sum_{n\in[N]} \frac{\boldsymbol{p}_n^\top}{N} & 0 \end{bmatrix}$$

$$= \begin{bmatrix} \mathrm{Cov}(\{\boldsymbol{p}_n\}_{n=1}^N) & (p+q-1)\boldsymbol{I} \\ (p+q-1)\boldsymbol{I} & 0 \end{bmatrix},$$

where $\mathrm{Cov}(\{\boldsymbol{p}_n\}_{n=1}^N) = \sum_{n\in[N]} \frac{1}{N} \left(\boldsymbol{p}_n - \sum_{n\in[N]} \frac{\boldsymbol{p}_n}{N}\right) \left(\boldsymbol{p}_n - \sum_{n\in[N]} \frac{\boldsymbol{p}_n}{N}\right)^\top$ is the covariance matrix of the set $\{\boldsymbol{p}_n\}_{n=1}^N$. Now we show that $\boldsymbol{M}$ cannot be positive definite. Suppose not. Then there exists a $\lambda > 0$ such that $\boldsymbol{v}^\top \boldsymbol{M} \boldsymbol{v} \geq \lambda \|\boldsymbol{v}\|^2$ for all $\boldsymbol{v} = (\boldsymbol{v}_1, \boldsymbol{v}_2) \in \mathbb{R}^{2d}$. This further implies that

$$\boldsymbol{v}_1^\top \mathrm{Cov}(\{\boldsymbol{p}_n\}_{n=1}^N) \boldsymbol{v}_1 + 2(p+q-1)\langle \boldsymbol{v}_1, \boldsymbol{v}_2 \rangle \geq \lambda \|\boldsymbol{v}\|^2, \quad \forall \boldsymbol{v}_1, \boldsymbol{v}_2 \in \mathbb{R}^d.$$

Taking $\boldsymbol{v}_1 = 0$ the above inequality implies that $\lambda \|\boldsymbol{v}_2\|^2 \leq 0$ for all $\boldsymbol{v}_2 \in \mathbb{R}^d$, which is a contradiction. Hence $\boldsymbol{M}$ cannot be positive definite and consequently neither can $\boldsymbol{H_\alpha}$. $\qquad\square$

## B.6 PROOF OF LEMMA 1

*Proof.* Consider the loss function $L(\cdot)$ given in Eq. (1). We can rewrite it as follows:

$$L(\bar{\boldsymbol{\theta}}) = -\frac{1}{N} \sum_{n\in[N]} \mathbb{E}_{x_1^{n+1}} \left[ x_{n+1} \cdot \log f_{\bar{\boldsymbol{\theta}}}(x_1^n) + (1 - x_{n+1}) \cdot \log(1 - f_{\bar{\boldsymbol{\theta}}}(x_1^n)) \right]$$

$$= -\frac{1}{N} \sum_{n\in[N]} \mathbb{E}_{x_1^n} \left[ \mathbb{E}_{x_{n+1}|x_1^n}[x_{n+1}] \cdot \log f_{\bar{\boldsymbol{\theta}}}(x_1^n) + \mathbb{E}_{x_{n+1}|x_1^n}[1 - x_{n+1}] \cdot \log(1 - f_{\bar{\boldsymbol{\theta}}}(x_1^n)) \right]$$

$$= -\frac{1}{N} \sum_{n \in [N]} \mathbb{E}_{x_1^n} \Big[ \mathbb{P}\left(x_{n+1} = 1 \mid x_n\right) \log \frac{f_{\bar{\boldsymbol{\theta}}}(x_1^n)}{\mathbb{P}\left(x_{n+1} = 1 \mid x_n\right)} + \mathbb{P}\left(x_{n+1} = 0 \mid x_n\right) \log \frac{1 - f_{\bar{\boldsymbol{\theta}}}(x_1^n)}{\mathbb{P}\left(x_{n+1} = 0 \mid x_n\right)}$$

$$- \mathbb{P}\left(x_{n+1} = 1 \mid x_n\right) \log \frac{1}{\mathbb{P}\left(x_{n+1} = 1 \mid x_n\right)} - \mathbb{P}\left(x_{n+1} = 0 \mid x_n\right) \log \frac{1}{\mathbb{P}\left(x_{n+1} = 0 \mid x_n\right)} \Big]$$

$$= -\frac{1}{N} \sum_{n \in [N]} \mathbb{E}_{x_1^n} \Big[ \mathbb{P}\left(x_{n+1} = 1 \mid x_n\right) \log \frac{f_{\bar{\boldsymbol{\theta}}}(x_1^n)}{\mathbb{P}\left(x_{n+1} = 1 \mid x_n\right)} + \mathbb{P}\left(x_{n+1} = 0 \mid x_n\right) \log \frac{1 - f_{\bar{\boldsymbol{\theta}}}(x_1^n)}{\mathbb{P}\left(x_{n+1} = 0 \mid x_n\right)} \Big]$$

$$- \mathbb{E}_{x_n} \Big[ \mathbb{E}_{x_1^{n-1} \mid x_n} \Big[ \mathbb{P}\left(x_{n+1} = 1 \mid x_n\right) \log \frac{1}{\mathbb{P}\left(x_{n+1} = 1 \mid x_n\right)} + \mathbb{P}\left(x_{n+1} = 0 \mid x_n\right) \log \frac{1}{\mathbb{P}\left(x_{n+1} = 0 \mid x_n\right)} \Big] \Big]$$

$$= -\frac{1}{N} \sum_{n \in [N]} \mathbb{E}_{x_1^n} \Big[ \mathbb{P}\left(x_{n+1} = 1 \mid x_n\right) \log \frac{f_{\bar{\boldsymbol{\theta}}}(x_1^n)}{\mathbb{P}\left(x_{n+1} = 1 \mid x_n\right)} + \mathbb{P}\left(x_{n+1} = 0 \mid x_n\right) \log \frac{1 - f_{\bar{\boldsymbol{\theta}}}(x_1^n)}{\mathbb{P}\left(x_{n+1} = 0 \mid x_n\right)} \Big]$$

$$- \mathbb{E}_{x_n} \Big[ \mathbb{P}\left(x_{n+1} = 1 \mid x_n\right) \log \frac{1}{\mathbb{P}\left(x_{n+1} = 1 \mid x_n\right)} + \mathbb{P}\left(x_{n+1} = 0 \mid x_n\right) \log \frac{1}{\mathbb{P}\left(x_{n+1} = 0 \mid x_n\right)} \Big].$$

Since $f_{\bar{\boldsymbol{\theta}}}(x_1^n) = \mathbb{P}_{\bar{\boldsymbol{\theta}}}\left(x_{n+1} = 1 \mid x_n\right)$, we have that the first term above is

$$\mathbb{P}\left(x_{n+1} = 1 \mid x_n\right) \log \frac{f_{\bar{\boldsymbol{\theta}}}(x_1^n)}{\mathbb{P}\left(x_{n+1} = 1 \mid x_n\right)} + \mathbb{P}\left(x_{n+1} = 0 \mid x_n\right) \log \frac{1 - f_{\bar{\boldsymbol{\theta}}}(x_1^n)}{\mathbb{P}\left(x_{n+1} = 0 \mid x_n\right)}$$
$$= -D_{\mathrm{KL}}(\mathbb{P}\left(x_{n+1} = \cdot \mid x_n\right) \parallel \mathbb{P}_{\bar{\boldsymbol{\theta}}}\left(x_{n+1} = \cdot \mid x_1^n\right)).$$

Further, observe that the second term is exactly the entropy rate $H(x_{n+1}|x_n)$. Hence, the above expression for the loss reduces to

$$L(\bar{\boldsymbol{\theta}}) = \frac{1}{N} \sum_{n \in [N]} \mathbb{E}_{x_1^n} \left[ D_{\mathrm{KL}}(\mathbb{P}\left(x_{n+1} = \cdot \mid x_n\right) \parallel \mathbb{P}_{\bar{\boldsymbol{\theta}}}\left(x_{n+1} = \cdot \mid x_1^n\right)) \right] + H(x_{n+1}|x_n),$$

and we are done. $\qquad\square$

### B.7 PROOF OF LEMMA 2

*Proof.* It suffices to show that for any component $\bar{\boldsymbol{\theta}}_j$ of $\bar{\boldsymbol{\theta}} \in \mathbb{R}^D$,

$$\frac{\partial}{\partial \bar{\boldsymbol{\theta}}_j} L(\bar{\boldsymbol{\theta}}) = -\frac{1}{N} \sum_{n \in [N]} \mathbb{E}_{x_1^{n+1}} \left[ (x_{n+1} - f_{\bar{\boldsymbol{\theta}}}(x_1^n)) \cdot \frac{\partial}{\partial \bar{\boldsymbol{\theta}}_j} \left( \boldsymbol{a}^\top \boldsymbol{z}_n + b \right) \right]$$

$$= -\frac{1}{N} \sum_{n \in [N]} \mathbb{E}_{x_1^n} \left[ (\mathbb{P}\left(x_{n+1} = 1 \mid x_n\right) - f_{\bar{\boldsymbol{\theta}}}(x_1^n)) \cdot \frac{\partial}{\partial \bar{\boldsymbol{\theta}}_j} \left( \boldsymbol{a}^\top \boldsymbol{z}_n + b \right) \right].$$

Recall from Eq. (3) that $L(\cdot)$ is given by

$$L(\bar{\boldsymbol{\theta}}) = -\frac{1}{N} \sum_{n \in [N]} \mathbb{E}_{x_1^{n+1}} \left[ x_{n+1} \cdot \log f_{\bar{\boldsymbol{\theta}}}(x_1^n) + (1 - x_{n+1}) \cdot \log(1 - f_{\bar{\boldsymbol{\theta}}}(x_1^n)) \right],$$

which implies that

$$\frac{\partial}{\partial \bar{\boldsymbol{\theta}}_j} L(\bar{\boldsymbol{\theta}}) = -\frac{1}{N} \sum_{n \in [N]} \mathbb{E}_{x_1^{n+1}} \left[ x_{n+1} \cdot \frac{\partial}{\partial \bar{\boldsymbol{\theta}}_j} \log f_{\bar{\boldsymbol{\theta}}}(x_1^n) + (1 - x_{n+1}) \cdot \frac{\partial}{\partial \bar{\boldsymbol{\theta}}_j} \log(1 - f_{\bar{\boldsymbol{\theta}}}(x_1^n)) \right]$$

$$= -\frac{1}{N} \sum_{n \in [N]} \mathbb{E}_{x_1^{n+1}} \left[ x_{n+1} \cdot \frac{1}{f_{\bar{\boldsymbol{\theta}}}(x_1^n)} \frac{\partial}{\partial \bar{\boldsymbol{\theta}}_j} f_{\bar{\boldsymbol{\theta}}}(x_1^n) + (1 - x_{n+1}) \cdot \frac{1}{1 - f_{\bar{\boldsymbol{\theta}}}(x_1^n)} \frac{\partial}{\partial \bar{\boldsymbol{\theta}}_j} (1 - f_{\bar{\boldsymbol{\theta}}}(x_1^n)) \right].$$

Since $f_{\bar{\boldsymbol{\theta}}}(x_1^n) = \sigma(\boldsymbol{a}^\top \boldsymbol{z}_n + b)$, we first note that the derivative of $\sigma$ is given by $\sigma'(z) = \frac{e^{-z}}{(1+e^{-z})^2} = \sigma(z)(1 - \sigma(z))$. Hence, the derivative $\frac{\partial}{\partial \bar{\boldsymbol{\theta}}_j} f_{\bar{\boldsymbol{\theta}}}(x_1^n)$ can be written as

$$\frac{\partial}{\partial \bar{\boldsymbol{\theta}}_j} f_{\bar{\boldsymbol{\theta}}}(x_1^n) = \frac{\partial}{\partial \bar{\boldsymbol{\theta}}_j} \sigma(\boldsymbol{a}^\top \boldsymbol{z}_n + b) = \sigma(\boldsymbol{a}^\top \boldsymbol{z}_n + b) \left[ 1 - \sigma(\boldsymbol{a}^\top \boldsymbol{z}_n + b) \right] \frac{\partial}{\partial \bar{\boldsymbol{\theta}}_j} (\boldsymbol{a}^\top \boldsymbol{z}_n + b)$$

$$= f_{\bar{\boldsymbol{\theta}}}(x_1^n)\big[1 - f_{\bar{\boldsymbol{\theta}}}(x_1^n)\big]\frac{\partial}{\partial\bar{\boldsymbol{\theta}}_j}(\boldsymbol{a}^\top \boldsymbol{z}_n + b).$$

Plugging this into the above expression, we have

$$\frac{\partial}{\partial\bar{\boldsymbol{\theta}}_j}L(\bar{\boldsymbol{\theta}}) = -\frac{1}{N}\sum_{n\in[N]}\mathbb{E}_{x_1^{n+1}}\left[x_{n+1}\cdot\big[1 - f_{\bar{\boldsymbol{\theta}}}(x_1^n)\big]\frac{\partial}{\partial\bar{\boldsymbol{\theta}}_j}(\boldsymbol{a}^\top \boldsymbol{z}_n + b) - (1 - x_{n+1})\cdot f_{\bar{\boldsymbol{\theta}}}(x_1^n)\frac{\partial}{\partial\bar{\boldsymbol{\theta}}_j}(\boldsymbol{a}^\top \boldsymbol{z}_n + b)\right]$$

$$= -\frac{1}{N}\sum_{n\in[N]}\mathbb{E}_{x_1^{n+1}}\left[x_{n+1}\cdot\big[1 - f_{\bar{\boldsymbol{\theta}}}(x_1^n)\big]\frac{\partial}{\partial\bar{\boldsymbol{\theta}}_j}(\boldsymbol{a}^\top \boldsymbol{z}_n + b) - (1 - x_{n+1})\cdot f_{\bar{\boldsymbol{\theta}}}(x_1^n)\frac{\partial}{\partial\bar{\boldsymbol{\theta}}_j}(\boldsymbol{a}^\top \boldsymbol{z}_n + b)\right]$$

$$= -\frac{1}{N}\sum_{n\in[N]}\mathbb{E}_{x_1^{n+1}}\left[(x_{n+1} - f_{\bar{\boldsymbol{\theta}}}(x_1^n))\cdot\frac{\partial}{\partial\bar{\boldsymbol{\theta}}_j}\left(\boldsymbol{a}^\top \boldsymbol{z}_n + b\right)\right]$$

$$= -\frac{1}{N}\sum_{n\in[N]}\mathbb{E}_{x_1^n}\left[(\mathbb{E}_{x_{n+1}|x_1^n}[x_{n+1}] - f_{\bar{\boldsymbol{\theta}}}(x_1^n))\cdot\frac{\partial}{\partial\bar{\boldsymbol{\theta}}_j}\left(\boldsymbol{a}^\top \boldsymbol{z}_n + b\right)\right]$$

$$= -\frac{1}{N}\sum_{n\in[N]}\mathbb{E}_{x_1^n}\left[(\mathbb{P}\left(x_{n+1} = 1 \mid x_n\right) - f_{\bar{\boldsymbol{\theta}}}(x_1^n))\cdot\frac{\partial}{\partial\bar{\boldsymbol{\theta}}_j}\left(\boldsymbol{a}^\top \boldsymbol{z}_n + b\right)\right],$$

and we are done. $\qquad\square$

## C  ADDITIONAL RESULTS FOR FIRST-ORDER MARKOV CHAINS

### C.1  MODEL ARCHITECTURE AND HYPER-PARAMETERS

Table 1: Parameters in the transformer architecture with their shape.

| Parameter | Matrix shape |
|---|---|
| transformer.wte | $2 \times d$ |
| transformer.wpe | $N \times d$ |
| transformer.h.ln_1 ($\times \ell$) | $d \times 1$ |
| transformer.h.attn.c_attn ($\times \ell$) | $3d \times d$ |
| transformer.h.attn.c_proj ($\times \ell$) | $d \times d$ |
| transformer.h.ln_2 ($\times \ell$) | $d \times 1$ |
| transformer.h.mlp.c_fc ($\times \ell$) | $4d \times d$ |
| transformer.h.mlp.c_proj ($\times \ell$) | $d \times 4d$ |
| transformer.ln_f | $d \times 1$ |

Table 2: Settings and parameters for the transformer model used in the experiments.

| | |
|---|---|
| Dataset | $k$-th order binary Markov source |
| Architecture | Based on the GPT-2 architecture as implemented in Pagliardini (2023) |
| Batch size | Grid-searched in $\{16, 50\}$ |
| Accumulation steps | 1 |
| Optimizer | AdamW ($\beta_1 = 0.9, \beta_2 = 0.95$) |
| Learning rate | 0.001 |
| Scheduler | Cosine |
| # Iterations | 8000 |
| Weight decay | $1 \times 10^{-3}$ |
| Dropout | 0 |
| Sequence length | Grid-searched in $\{512, 1024, 2048\}$ |
| Embedding dimension | Grid-searched in $\{4, 8, 16, 32, 64\}$ |
| Transformer layers | Between 1 and 6 depending on the experiment |
| Attention heads | Grid-searched in $\{1, 2, 4, 8\}$ |
| Mask window | Between 2 and full causal masking depending on the experiment |
| Repetitions | 3 or 5 |

## D  EMPIRICAL FORMULA FO $p + q < 1$ BASED ON LOW-RANK SOLUTIONS

In this section we compute the function $f_{\boldsymbol{\theta}}(x_1^n)$ that gives the next-symbol probability predicted by the network, using the values of the weight matrices obtained five independent experiment runs. By substituting the empirical weights into the transformer architecture from § A, i.e.

$$\boldsymbol{x}_n = x_n \, \boldsymbol{e} + \boldsymbol{p}_n \in \mathbb{R}^d, \qquad \text{(Uni-embedding)}$$

$$\boldsymbol{y}_n = \boldsymbol{x}_n + \boldsymbol{W}_O \sum_{i \in [n]} \text{att}_{n,i} \cdot \boldsymbol{W}_V \, \boldsymbol{x}_i \in \mathbb{R}^d, \qquad \text{(Attention)}$$

$$\boldsymbol{z}_n = \boldsymbol{y}_n + \boldsymbol{W}_2 \, \text{ReLU}(\boldsymbol{W}_1 \, \boldsymbol{y}_n) \in \mathbb{R}^d, \qquad \text{(FF)}$$

$$\text{logit}_n = \langle \boldsymbol{a}, \boldsymbol{z}_n \rangle + b \qquad \in \mathbb{R}, \qquad \text{(Linear)}$$

$$f_{\boldsymbol{\theta}}(x_1^n) \triangleq \mathbb{P}_{\boldsymbol{\theta}} \left( x_{n+1} = 1 \mid x_1^n \right) = \sigma(\text{logit}_n). \qquad \text{(Prediction)}$$

We can obtain an explicit expression for $f_{\boldsymbol{\theta}}(x_1^n)$ as it is actually learned by the model. We now analyze each section of the model architecture separately.

**Embedding.** All the five independent runs show that the word embedding vector $\boldsymbol{e}$ has the structure

$$\boldsymbol{e} = e \cdot \boldsymbol{v} \tag{15}$$

where $\boldsymbol{v} = (v_1, \ldots, v_d)$ is such that $v_i \in \{-1, +1\}$ for all $i$, i.e., $\boldsymbol{v} \in \{-1, 1\}^d$, and $e$ is some constant. Moreover, the positional embeddings are approximately constant across positions $n$, and they share a similar structure to $\boldsymbol{e}$. In particular, we always have that

$$\boldsymbol{p}_n = \boldsymbol{p} = p \cdot \boldsymbol{v} \quad \forall n \tag{16}$$

for some constant $p \in \mathbb{R}$. Furthermore, the constants are always such that $p < 0$ and $e + p > 0$.

**Attention.** Across all the runs, we observe that the contribution of the attention mechanism is negligible compared to the skip-connection. In particular, we observe that

$$\frac{\|\boldsymbol{W}_O \sum_{i \in [n]} \mathrm{att}_{n,i} \cdot \boldsymbol{W}_V \boldsymbol{x}_i\|}{\|\boldsymbol{y}_n\|} \approx 0.01 \tag{17}$$

uniformly for all $n$. Therefore, we can use the approximation

$$\boldsymbol{y}_n \approx \boldsymbol{x}_n \quad \forall n. \tag{18}$$

**FF.** For the MLP layer, we observe that $\boldsymbol{W}_1$ and $\boldsymbol{W}_2$ have a clear joint structure. In fact, we empirically see that

$$\boldsymbol{W}_1 = w_1 \cdot \boldsymbol{w} \cdot \boldsymbol{v}^T \tag{19}$$

where $\boldsymbol{v}$ is again the same vector as in Eq. (15), $\boldsymbol{w} \in \{-1, 1\}^r$ and $w_1 \in \mathbb{R}$. Hence, $\boldsymbol{W}_1$ is a rank-one matrix. As customary in the GPT-2 model, for our experiments we used $r = 4d = 16$. Furthermore, we see that

$$\boldsymbol{W}_2 = \boldsymbol{W}_1^T. \tag{20}$$

Due to this structure and the formula for $\boldsymbol{y}_n$ described above, we have

$$\boldsymbol{W}_1 \boldsymbol{y}_n = \boldsymbol{W}_1 \boldsymbol{x}_n = w_1 d(ex_n + p)\boldsymbol{w} \tag{21}$$

Let now $\boldsymbol{r} = \mathrm{ReLU}(\boldsymbol{W}_1 \boldsymbol{y}_n)$. Due to the fact that $p < 0$ and $e + p > 0$, we have that, if $x_n = 1$,

$$r_i = \begin{cases} e + p, & \text{if } w_i = 1, \\ 0, & \text{if } w_i = -1. \end{cases} \tag{22}$$

While if $x_n = 0$,

$$r_i = \begin{cases} 0, & \text{if } w_i = 1, \\ -p, & \text{if } w_i = -1. \end{cases} \tag{23}$$

Let $\beta = \sum_{i=1}^r \mathbb{1}_{\{w_i=1\}}$. Since $\boldsymbol{W}_2 = \boldsymbol{W}_1^T = w_1 \boldsymbol{v} \cdot \boldsymbol{w}^T$, we have that, for $\widetilde{\boldsymbol{r}} = \boldsymbol{W}_2 \boldsymbol{r}$,

$$\widetilde{\boldsymbol{r}} = \begin{cases} w_1^2 d(e + p)\beta \cdot \boldsymbol{v}, & \text{if } x_n = 1, \\ w_1^2 dp(r - \beta) \cdot \boldsymbol{v}, & \text{if } x_n = 0. \end{cases} \tag{24}$$

Or more compactly,

$$\widetilde{\boldsymbol{r}} = w_1^2 d(ex_n + p)((2\beta - r)x_n + r - \beta) \cdot \boldsymbol{v}, \tag{25}$$

and

$$\boldsymbol{z}_n = \boldsymbol{y}_n + \widetilde{\boldsymbol{r}} = (ex_n + p)(1 + w_1^2 d((2\beta - r)x_n + r - \beta)) \cdot \boldsymbol{v} \tag{26}$$

**Linear.** Since $\boldsymbol{a} = \boldsymbol{e}$ due to weight-tying, we have

$$\mathrm{logit}_n = ed(ex_n + p)(1 + w_1^2 d((2\beta - r)x_n + r - \beta)) + b \tag{27}$$

**Prediction.** We can now plug in the empirical values obtained by averaging five independent runs. The numerical results that we get are

$$\begin{aligned} e &= 0.3618 \\ p &= -0.1539 \\ w_1 &= 0.3264 \\ b &= -0.1229 \\ \beta &= 5 \end{aligned} \tag{28}$$

Plugging these numbers into Eq. (27), we get

$$\text{logit}_n = \begin{cases} 0.8191, & \text{if } x_n = 1, \\ -1.3897, & \text{if } x_n = 0. \end{cases} \tag{29}$$

Hence, by applying the sigmoid function to the logit values, we obtain the predicted probabilities

$$f_{\boldsymbol{\theta}}(x_1^n) = \mathbb{P}_{\boldsymbol{\theta}}\left(x_{n+1} = 1 \mid x_n\right) = \begin{cases} \sigma(0.8191) = 0.694, & \text{if } x_n = 1, \\ \sigma(-1.3897) = 0.199, & \text{if } x_n = 0. \end{cases} \tag{30}$$

The numerical results correspond almost exactly to the expected theoretical values of $1 - q = 0.7$ and $p = 0.2$.

