# OpenReview forum: "Attention with Markov: A Curious Case of Single-layer Transformers"
_ICLR.cc/2025/Conference — ICLR 2025 Spotlight_

### Official Review · Reviewer_FHWC · 2024-10-24

**Soundness:** 3
**Presentation:** 3
**Contribution:** 2
**Rating:** 6
**Confidence:** 3

**Summary:**

This paper introduces a new framework for analyzing transformers through the lens of Markov chains, which is a novel and theoretically rich approach. This framework enhances our understanding of how transformers handle sequential data, particularly first-order Markov processes. Specifically, the theoretical characterization of the loss landscape for single-layer transformers, identifying the presence of both global minima (bigram) and bad local minima (unigram), is a good contribution. This helps explain the empirical observations and provides a solid foundation for future work.

**Strengths:**

+ The theoretical results are novel and informative.

+ The experiments are well-designed, providing solid support of the considered insights.

**Weaknesses:**

I found no explicit assumptions in the paper, which may imply that few assumptions are utilized in the theoretical analysis. Thus, I suggest that the authors list the assumptions that were used.

The set of words under consideration is too toyish, which is the primary concern.

**Questions:**

Could the authors clarify the difference between the following work [1] and the proposed results?

[1] Training Dynamics of Multi-Head Softmax Attention for In-Context Learning: Emergence, Convergence, and Optimality. COLT, 2024.

---

> ### Author Response · Authors · 2024-11-21
> **Assumptions and extensions to multi-state Markov are already highlighted in the paper**
>
> We thank the reviewer for the encouraging feedback and insightful comments. We address the individual questions below.
>
> **Assumptions:** We would like to highlight that Section 3 in the paper already delineates the assumptions behind our current theoretical results (Theorems 1-3). In particular, they are:
>
>   - **1. Data:** We assume that the input sequence is a binary Markov chain with state transition probabilities $p$ and $q$, as illustrated in Figure 2b of the paper.
>
> - **2. Model:** We consider a single-layer transformer with single-head attention, without layer norm. For the input embedding, without loss of generality, we consider the (Uni-embedding) structure in Line 213 in the paper. The rest of the architecture is the same as the one illustrated in Figure 2a and Lines 124-131 in the paper.  For this architecture, we study both the weight-tied and non weight-tied settings in Sections 4.1 and 4.2 respectively.
>
> **Vocabulary size:** Please note that while our current theoretical results are for the binary setting, they readily generalize to the multi-state scenario as discussed in Section 4.3 in the paper. This demonstrates the generality of our results and observations.
>
> **Comparison to [1]:** The suggested work [1] on ICL for single-layer transformers differs fundamentally from our setting. While [1] examines gradient flow dynamics in a single-layer transformer, their focus is on a linear regression task, which is markedly distinct from our sequential learning task with Markovian inputs. Therefore, a direct comparison between our results and theirs is not applicable.
>
> -------------------
> **References**
>
> [1] Training Dynamics of Multi-Head Softmax Attention for In-Context Learning: Emergence, Convergence, and Optimality. COLT, 2024.

---

> > ### Author Response · Authors · 2024-12-02
> > **Discussion period ending tomorrow**
> >
> > Dear Reviewer FHWC,
> >
> > We sincerely appreciate the time you have taken to provide valuable feedback for our work. As we are getting closer to the end of the discussion period, could you let us know if our responses above have adequately addressed your concerns? We remain at your disposal for any further questions.
> >
> > If you agree that our responses to your reviews have addressed the concerns you listed, we kindly ask that you consider whether raising your score would more accurately reflect your updated evaluation of our paper. Thank you again for your time and thoughtful comments!
> >
> > Sincerely, The Authors

---

> > > ### Comment · Reviewer_FHWC · 2024-12-03
> > > **Official Comments**
> > >
> > > Hi,
> > >
> > > Thanks for the detailed responses. Since my concerns are addressed, I will raise the score to 6.

---

> ### Author Response · Authors · 2024-12-01
>
> Dear Reviewer FHWC,
>
> We sincerely appreciate the time you have taken to provide valuable feedback for our work. As we are getting closer to the end of the discussion period, could you let us know if our responses above have adequately addressed your concerns? We remain at your disposal for any further questions.
>
> Sincerely, The Authors

---

### Official Review · Reviewer_dP9v · 2024-10-30

**Soundness:** 3
**Presentation:** 3
**Contribution:** 2
**Rating:** 5
**Confidence:** 4

**Summary:**

This paper introduces a framework for learning a first-order Markov chain using a single-layer transformer. Within this framework, the authors characterize the network’s loss landscape, identifying the presence of both global and bad local minima.

**Strengths:**

1. Examining the limitations of a single-layer transformer in learning a simple Markov kernel is an intriguing aspect of this work.

2. The authors characterize the roles of Markov switching probabilities and weight-tying in contributing to the presence of local optima.

3. Extensive experimental results are provided to support the theoretical insights

**Weaknesses:**

The main limitation of this paper is that the current results may not fully address their central question: systematically characterizing the learning gap of a single-layer transformer. The authors show that a bad local minimum exists when $p+q > 1$, but they do not rule out the possibility that only global optima exist when $p+q < 1$. Given that the results are primarily existence arguments, this does not entirely support the empirical observation that the transformer converges to a bad minimum in the $p+q > 1$ regime while learning the global minimum in the $p+q < 1$ regime. This implicit bias would benefit from additional theoretical support, such as insights from learning dynamics.  Moreover, the data model is overly simplified, which raises expectations for a more rigorous analysis that fully characterizes the solution learned by transformers rather than relying on existence arguments alone.

**Questions:**

1. See weakness.

2. In the proof sketch of Theorem 2 (line 363), the authors set $\mathbf{e} = \mathbf{a} = 0$, which implies that for the uni-embedding (line 213), $\mathbf{x}_n$ depends solely on the position encoding and lacks information from the state $x_n$. If I understand correctly, this setup could lead to problematic solutions for the model, as it is unable to fully utilize input information—potentially independent of the effects of weight tying. Could the authors clarify this point?

---

> ### Author Response · Authors · 2024-11-21
> **Loss landscape and learning dynamics**
>
> We thank the reviewer for the helpful feedback and insightful comments. We address the individual concerns below.
>
> **Characterizing the learning gap:** This is an excellent question. The primary motivation behind our work is to demystify the curious empirical behavior of single-layer transformers when trained on Markov chains. This is what we precisely characterize in our paper through the loss landscape results in Theorems 1-3, corroborating the empirical observations in Figure 3. This can be viewed as a static explanation of these phenomena through the loss landscape. With regards to the learning dynamics and the presence of local/global minima for $p+q \lessgtr 1$, the following are two key points:
>
>  - **(1) Existence results and what we empirically observe:** Although our theorems provide existence arguments, they align closely with empirical solutions for both $p+q > 1$ and $p+q < 1$. For $p+q > 1$, as discussed in Lines 370–405, the transformer model empirically converges to local minima that nearly replicate the theoretical construction in Theorem 2. Similarly, for $p+q < 1$, the global minima described in Theorem 1 perfectly match the empirically observed minima, as shown in Lines 264–295 and Appendix D. Together, these results confirm that our theoretical constructions rigorously characterize the solutions learned by transformers.
>
>  - **(2) Learning dynamics and why we observe above phenomena:** Indeed, as noted in Line 407, understanding why the model converges to bad minima for $p+q > 1$ but not for $p+q < 1$ requires analyzing its learning dynamics. A follow-up work [1] addresses this directly and identifies the key reason behind this phenomenon. Specifically, as shown in Figure 1 of [1],  the standard Gaussian initialization around zero falls in the basin of attraction for local minima when $p+q>1$, causing the model to get stuck at these minima. Conversely, for $p+q < 1$, the same initialization falls within the basin of attraction for global minima, enabling convergence to global minima.
>
> In other words, our results can be viewed as a static landscape picture of the curious phenomena exhibited by single-layer transformers, whereas [1] complements it through learning dynamics.
>
> **Line 363, proof sketch of Theorem 2**: Indeed, when $\boldsymbol{e} = \boldsymbol{a} = 0$, the model ignores input statistics entirely, yielding a constant prediction probability, as shown empirically in Figure 3c. While this holds regardless of weight-tying, Theorem 2 demonstrates that such solutions correspond to bad local minima with weight-tying, whereas they become saddle points without weight-tying (Theorem 3). This distinction arises from the impact of weight-tying on the Hessian at these critical points, as detailed in the proof of Theorem 2 (Appendix B.4).
>
> -----------------------------------------------
> **References**
>
> [1] Local to Global: Learning Dynamics and Effect of Initialization for Transformers, https://arxiv.org/abs/2406.03072

---

> > ### Comment · Reviewer_dP9v · 2024-11-25
> > **Main concern remains unaddressed**
> >
> > I appreciate the authors' efforts in addressing my concerns, but my main point remains:
> >
> > The alignment between empirical solutions and theoretical constructions is compelling. However, it is important to note that the motivation for this work stems from empirical observations. While it is possible to construct a specific theoretical setting that aligns with these empirical solutions, the theoretical results **do not provide further insight into why this gap occurs in practice** and **do not explain why or how the model transitions—or fails to transition—between these solutions**.
> >
> > As the authors acknowledge, addressing this gap requires a dynamics-based analysis. Without such an analysis, the current contribution feels incomplete and does not sufficiently address the problem outlined in the abstract and introduction, which risks overselling the paper’s impact. Therefore, I will maintain my score.

---

> > > ### Author Response · Authors · 2024-12-02
> > > **Loss landscape is as important as learning dynamics and is a foundational step for dynamics analysis**
> > >
> > > We thank the reviewer for their prompt response and explaining their rationale.
> > >
> > > While we agree that dynamics-based analysis is important to understand the **why** part of the curious phenomenon, we strongly believe that characterizing the **what** part is also equally important, which is the focus of our paper. A precise understanding of the loss landscape is a foundational step for any comprehensive gradient-dynamics analysis, making our thorough investigation a necessary first step. Given the depth and breadth of our results on the loss landscape, we chose to focus exclusively on this aspect in the current paper for the sake of clarity and brevity. This approach also lays a clear foundation for future dynamics-based analysis, as demonstrated in the concurrent work of [1].

---

### Official Review · Reviewer_rH7j · 2024-11-03

**Soundness:** 4
**Presentation:** 4
**Contribution:** 4
**Rating:** 10
**Confidence:** 3

**Summary:**

This paper establishes a general and novel framework for the theoretical and empirical study of the attention mechanism on Markov data. They theoretically show that for one layer of Transformers, if the data comes from a Markov chain with binary states and transition probability given by P = (1-p, p \\ q, 1-q), the global minima of the next-token cross-entropy loss can recover the optimal risk. They theoretically show that there exist bad local minima with weight tying if p+q > 1, and without weight tying, local minima become saddle point. Empirically, they validate the theory and show those saddle point can be avoided if without weight tying. They also present experimental results on multi-state Markov chain and mention some open problems about deep Transformers.

I really like this paper. I think the framework is novel, interesting, and worth investigating. The theoretical and empirical results are solid, and the presentation is particularly clear. Although this is very initial research, they propose some future directions about why single-layer Transformers can sometimes be trapped in bad local minima when trained on the Markov data, and why deep Transformers can behave well and avoid local minima or saddle point. I think we should encourage this style of theoretical research since there are too much theoretical research with complex setups and heavy math, that explains nothing at all.

**Strengths:**

1. The theoretical results are solid and can be validated via experiments.

2. They establish a novel framework to investigate the attention mechanism on Markov data.

3. The presentation is particularly clear and comfortable.

**Weaknesses:**

Current, I do not see any big issues.

**Questions:**

/

**Details Of Ethics Concerns:**

/

---

> ### Author Response · Authors · 2024-11-21
> **Thank you!**
>
> We sincerely thank the reviewer for their thoughtful evaluation and for the strong appraisal of our results and contributions. With many interesting research directions emanating along this theme, your positive feedback is a definite encouragement to continue this line of research.

---

### Official Review · Reviewer_pFAs · 2024-11-04

**Soundness:** 4
**Presentation:** 4
**Contribution:** 3
**Rating:** 8
**Confidence:** 4

**Summary:**

The authors study a curious case of transformers via Markov Chains, in which one-layer transformer struggles to find the global minimum in the Markov task. They further provide theoretical and empirical studies, showing that weight tying can lead to bad local minima when Markovian switching is greater than one. They also extend the results to multi-state Markov Chains. Experiments in multi-layer transformers further show that they can reach the global minimum, which is an intriguing future direction.

**Strengths:**

1. The problem setup is interesting and concise.
2. The presentation is clear.
3. The analysis is solid, containing both theory and extensive experiments.

**Weaknesses:**

The attention structure is missing here. Specifically, the Markov chain requires no attention structure, nor does theoretical analysis include the attention layer. It is more like "MLPs with Markov" instead of "attention with Markov".

**Questions:**

1. Can authors comment on the role of attention layers in the paper?
2. Can authors present the attention patterns of the transformers trained the Markov tasks?
3. If time permits, can authors train pure MLP layers on the Markov task?

---

> ### Author Response · Authors · 2024-11-22
> **Role of attention mechanism**
>
> We thank the reviewer for their strong appraisal and encouraging comments about our work. We address the individual questions below.
>
> **Role of attention mechanism:** Indeed, surprisingly, it turns out that a single-layer transformer can effectively model a first-order Markov chain without relying heavily on the attention mechanism, leveraging the skip connection instead. Note that this behavior arises naturally from the inherent structure of the transformer architecture itself and the input being first-order, not from any explicit design on our part. While the diminished reliance on attention might initially seem uninteresting, what is striking is that the model automatically learns to disregard attention during gradient-based training. Our experimental results validate this theoretical insight, as evidenced by the approximation error between the attention output $\boldsymbol{y}_n$ and skip-connection $\boldsymbol{x}_n$, and the underlying attention patterns visualized here: Figures 1 and 2 of https://anonymous.4open.science/r/Markov-6B43/Rebuttal/iclr-rebuttal.pdf
>
> On the other hand, if we increase the depth of the transformer, it starts exhibiting in-context learning (ICL) capabilities when trained on Markovian inputs. Here, the attention layer plays a key role in realizing the induction-head mechanism, a cornerstone component behind ICL. The following works study this in detail: https://arxiv.org/abs/2407.17686v1, https://arxiv.org/abs/2402.14735
>
> **Pure MLP layers:** Thank you for your question! If you're asking about training the transformer with the attention layer removed but keeping everything else, including the MLP, intact, our results suggest that the model would indeed succeed in learning the Markovian kernel. This setup simplifies the task compared to the current scenario. In the present setting, even though the attention layer is included, the model eventually learns to disregard it. Removing the attention layer outright would make it easier for the model to directly learn the first-order kernel. Figure 3 here indeed confirms this observation: https://anonymous.4open.science/r/Markov-6B43/Rebuttal/iclr-rebuttal.pdf

---

### Meta-Review · Area_Chair_QD6e · 2024-12-20

**Metareview:**

This paper introduces a theoretical and empirical framework to understand how single-layer transformers learn (or fail to learn) first-order Markov chains.

After carefully reviewing the paper and discussion, the AC believes that most major concerns have been resolved, and the remaining concerns are minor. The finding that single-layer transformers can get stuck at local minima when modeling first-order Markov chains could be valuable for future work. Therefore, the AC recommends accepting this paper.

The authors should carefully integrate all the discussions and add more emphasis on why the finding is important in the final version to further highlight the contribution.

**Additional Comments On Reviewer Discussion:**

This paper demonstrates that single-layer transformers can get stuck at local minima corresponding to a unigram distribution under certain conditions, whereas global minima representing the bigram distribution exist. The authors delineate conditions (such as the choice of model parameters and the presence of weight tying) that lead to either favorable global minima or unfavorable local minima.

During the rebuttal and discussion phase, several concerns were raised by the reviewers and addressed by the authors. These concerns included: Practical Insight (*Reviewer dP9v*), Clarity of Theoretical Results and Assumptions (*Reviewer FHWC*), Attention Mechanism and Comparison to Related Work (*Reviewer pFA* and *Reviewer FHWC*).

There remains a concern about the theoretical contribution, as acknowledged by *Reviewer dP9v*: "The theoretical results only align with part of the empirical observations but cannot explain and do not provide deeper insights..." However, considering that the finding that single-layer transformers could potentially be valuable for and motivate future works, such as graph representation learning and causal discovery, the AC believes that this concern is not significant enough to warrant rejecting this paper.

---

### Decision · Program_Chairs · 2025-01-22

Accept (Spotlight)